# Interaction between SNAI2 and MYOD enhances oncogenesis and suppresses differentiation in Fusion Negative Rhabdomyosarcoma

Silvia Pomella[1,2,11], Prethish Sreenivas [3,11], Berkley E. Gryder [2,11], Long Wang[3], David Milewski [2], Matteo Cassandri[1], Kunal Baxi[3], Nicole R. Hensch [3], Elena Carcarino[1], Young Song[2], Hsien-Chao Chou [2], Marielle E. Yohe[2,4], Benjamin Z. Stanton [5], Bruno Amadio[6], Ignazio Caruana [1], Cristiano De Stefanis[7], Rita De Vito[8], Franco Locatelli[1,9], Yidong Chen[3], Eleanor Y. Chen[10], Peter Houghton[3], Javed Khan [2,12✉], Rossella Rota [1,12✉] & Myron S. Ignatius [3,12✉]

Rhabdomyosarcoma (RMS) is an aggressive pediatric malignancy of the muscle, that includes Fusion Positive (FP)-RMS harboring PAX3/7-FOXO1 and Fusion Negative (FN)-RMS commonly with RAS pathway mutations. RMS express myogenic master transcription factors MYOD and MYOG yet are unable to terminally differentiate. Here, we report that SNAI2 is highly expressed in FN-RMS, is oncogenic, blocks myogenic differentiation, and promotes growth. MYOD activates *SNAI2* transcription via super enhancers with striped 3D contact architecture. Genome wide chromatin binding analysis demonstrates that SNAI2 preferentially binds enhancer elements and competes with MYOD at a subset of myogenic enhancers required for terminal differentiation. SNAI2 also suppresses expression of a muscle differentiation program modulated by *MYOG*, *MEF2*, and *CDKN1A*. Further, RAS/MEK-signaling modulates SNAI2 levels and binding to chromatin, suggesting that the differentiation blockade by oncogenic RAS is mediated in part by SNAI2. Thus, an interplay between SNAI2, MYOD, and RAS prevents myogenic differentiation and promotes tumorigenesis.

[1] Department of Pediatric Hematology and Oncology, Bambino Gesù Children's Hospital, IRCCS, Rome, Italy. [2] Genetics Branch, NCI, NIH, Bethesda, MD, USA. [3] Greehey Children's Cancer Research Institute, Department of Molecular Medicine, University of Texas Health Sciences Center, San Antonio, Texas, USA. [4] Pediatric Oncology Branch, NCI, NIH, Bethesda, MD, USA. [5] Center for Childhood Cancer and Blood Diseases, Abigail Wexner Research Institute at Nationwide Children's Hospital, The Ohio State University, Columbus, OH 43205, USA. [6] SAFU Laboratory, Translational Research Area, Regina Elena National Cancer Institute, Rome, Italy. [7] Histology-Core Facility, Bambino Gesu' Children's Hospital, IRCCS, Rome, Italy. [8] Department of Pathology Unit, Department of Laboratories, Bambino Gesu' Children's Hospital, IRCCS, Rome, Italy. [9] Departmentof Pediatrics, Sapienza University of Rome, Rome, Italy. [10] Department of Pathology, University of Washington, Seattle, WA 98195, USA. [11]These authors contributed equally: Silvia Pomella, Prethish Sreenivas, Berkley E. Gryder. [12]These authors jointly supervised this work: Javed Khan, Rossella Rota, Myron S. Ignatius. ✉email: khanjav@mail.nih.gov; rossella.rota@opbg.net; ignatius@uthscsa.edu

Rhabdomyosarcoma (RMS) is a pediatric malignancy of the muscle that accounts for 50% of all soft tissue sarcomas in childhood and 7–8% of all pediatric cancers. RMS is best described as a tumor of skeletal muscle lineage blocked in various stages of differentiation. The two common histological subtypes, Alveolar or Embryonal RMS, are now molecularly defined by the presence (Fusion Positive; FP-RMS) or absence (Fusion Negative; FN-RMS) of PAX3/7-fusion genes respectivley[1,2]. In FP-RMS, the most common lesion is the PAX3/7-FOXO1 gene fusion, while FN-RMS most frequently harbor RAS pathway mutations[3,4]. Current multimodal therapy includes a combination of surgery, radiotherapy, and chemotherapy[5,6]; but despite this, the survival rate in metastatic refractory RMS is less than 30%[7,8]. Therefore, there is a need to better understand the molecular underpinnings of RMS, to develop novel therapies, and reduce toxicity thereby improving morbidity and mortality due to this aggressive disease.

RMS cells robustly express the myogenic transcription factors MYOD and MYOG, but yet are unable to terminally differentiate. Competition between MYOD and HEY1, TWIST1 and splice-forms of co-factor E2A have been shown to interfere with MYOD transcriptional activity and block terminal differentiation, however, this does not explain how MYOD is able to drive early myogenic gene expression (MYOG), yet unable to mediate terminal differentiation[9–11].

In this study, we show that SNAI2, a member of the SNAIL family of transcriptional factors, is highly expressed in FN-RMS and acts as an oncogene through selective inhibition of MYOD-driven differentiation and by down-regulating multiple transcription factors necessary for terminal differentiation, thereby blocking myogenic differentiation and supporting growth. These effects cannot be solely ascribed to SNAI2 functions in the epithelial to mesenchymal transition[12], but rather to fundamental effects on muscle biology. SNAI2 is known to repress the transcription of target genes by specifically binding the consensus motif CANNTG (i.e., E-box), and there are only a few reports characterizing genome-wide SNAI2 binding but none in RMS[13–15]. Here we report that SNAI2 directly represses the expression of MYOD bound differentiation genes, including MYOG, MEF2A, and TNNI1, by binding to E-box motifs mostly at genic enhancers. We define several critical aspects of SNAI2 functioning as an oncogene in FN-RMS. First, we define the SNAI2 regulatory locus and identify MYOD as a direct regulator of SNAI2 expression. Second, we connect mechanistically the cellular and molecular effects of SNAI2 on blocking tumor differentiation and promoting growth. Additionally, we show that combining SNAI2 reduction with standard of care vincristine results in synergistic reduction in tumor volume and robust enhancement of muscle differentiation. Lastly, we report convergent transcriptional consequences from either inhibition of mutant RAS signaling or depletion of SNAI2, suggesting a functional link between SNAI2 and the RAS pathway in FN-RMS.

## Results

### SNAI2 is highly expressed in RMS and is regulated by MYOD bound super enhancers.
We first investigated expression of SNAI2 and found it is up-regulated in tumor samples and cell lines compared to normal tissue with FN-RMS having higher expression by RNA-seq analysis (>4log$_2$ FPKM) (Fig. 1a)[4]. In RMS cell lines the higher mRNA expression was associated with higher SNAI2 protein levels when compared to primary human myoblasts (Fig. 1b). Of note, SNAI2 is located on chromosome 8 which is gained in two thirds of all RMS tumors[4]. Analysis of SNAI2 expression in a different cohort of approximately 2000 pediatric cancers from the St. Jude-PeCan portal confirmed that SNAI2 is highly expressed in RMS tumors and, especially, FN-RMS compared to other pediatric cancers

with only osteosarcoma tumors expressing higher SNAI2 (Fig. S1a). Immunohistochemical staining also revealed high protein expression of SNAI2 in 14 of 19 FN-RMS and 3 of 4 FP-RMS pediatric primary tumors (Fig. 1c and Fig. S1b).

Tumor-type or cell-type selective expression of a gene is controlled often by nearby super enhancers (SEs) that are only active in the relevant cellular state[16]. To establish the genomic loci within which SNAI2 enhancers may exist, we defined the tridimensional structure around the SNAI2 locus by analyzing Hi-C data reported for human lung fibroblasts[17] and found a "stripe" pattern[18] of 1.2 Mb within a topologically associated domain (TAD) containing SNAI2 (Fig. 1d), indicating high levels of interactions, and spatial organization. Multiple active enhancers and SEs were discovered within this SNAI2 TAD using the H3K27ac mark in a set of 11 FP-RMS (6 cell lines and 5 primary tumors) and 9 FN-RMS (5 cell lines and 4 primary tumors)[19] (Fig. 1e). Many active SEs in human skeletal myoblasts strongly overlap with RMS SEs (both in FP-RMS and FN-RMS), which are reduced during differentiation[19]. Similarly, we find that the SNAI2 SEs that are maintained in RMS are among those that decrease during myogenesis and are mostly absent in fully mature muscle (Fig. 1e).

The higher expression of SNAI2 compared to normal tissues and its expression in a majority of RMS samples suggested potential control by master regulators known to be shared between both FP-RMS and FN-RMS[19,20]. Myod through binding to a proximal promoter region can regulate Snai2 expression during muscle regeneration and in the early phase of differentiation in murine myoblasts[21]. To further investigate the transcriptional regulation of SNAI2 by MYOD in FN-RMS, we first used ChIP-seq to assess MYOD binding in mutant RAS-driven FN-RMS cells (SMS-CTR and RD)[22,23]. We found that the SNAI2 cis-regulatory environment is bound by MYOD in active chromatin (H3K27ac), not only at the proximal promoter but also at distal enhancer and SE sites (Fig. 1f). Examining the same sites in myoblasts and myotubes revealed that MYOD and H3K27ac are present but decrease at SNAI2 sites during muscle differentiation (Fig. 1f). To evaluate the chromatin conformation around the SNAI2 locus in FN-RMS, we performed HiChIP in SMS-CTR cells (Fig. 1g). We confirmed that in FN-RMS SNAI2 regulatory regions interact along a "stripe" that includes the super enhancers E1–E3. Expression analysis of MYOD1 and SNAI2 revealed a significant positive correlation in RMS and normal muscle but not in non-muscle tissues (FN-RMS: Pearson 0.428, $p = 0.0000038$; FP-RMS: Pearson 0.372, $p = 0.0001$; Muscle: Pearson 0.494, $p = 0.001$; Other: Pearson 0.051, $p = 0.477$) (Fig. S1c). These observations led us to investigate if SNAI2 expression depends on MYOD in FN-RMS. Utilizing short interference RNA (siRNA) knockdown of MYOD, we assessed SNAI2 expression in three FN-RMS cell lines (RD, JR1, and SMS-CTR). We found that MYOD knockdown in all three cell lines suppressed SNAI2 expression at both protein and mRNA levels at 24 and 48 h post-transfection (Fig. 1h and Fig. S1d). Conversely, when MYOD is ectopically expressed in human mesenchymal-derived fibroblasts lacking MYOD, SNAI2 expression is up-regulated[24] (Fig. S1e). Analysis of a published model of MYOD1 induction by PAX3-FOXO1 in fibroblasts[19] (Fig. S1f, upper) revealed a focused increase of H3K27ac at MYOD sites corresponding to those in SMS-CTR cells on the SNAI2 locus, along with an increase in SNAI2 expression (Fig. S1f, lower).

To test the direct requirement of MYOD binding sites to regulate SNAI2, we performed CRISPRi via dCas9-KRAB recruited to MYOD motifs on the SNAI2 locus in SMS-CTR cells (Fig. 1i). After 72 h of guide RNA mediated repression of 5 candidate regions (E1–E5), we find that SNAI2 transcription was strongly dependent on two binding sites with the highest MYOD deposits (E1 and E2) but not on the more distant sites (E3–E5) with the lowest MYOD signal (Fig. 1i).

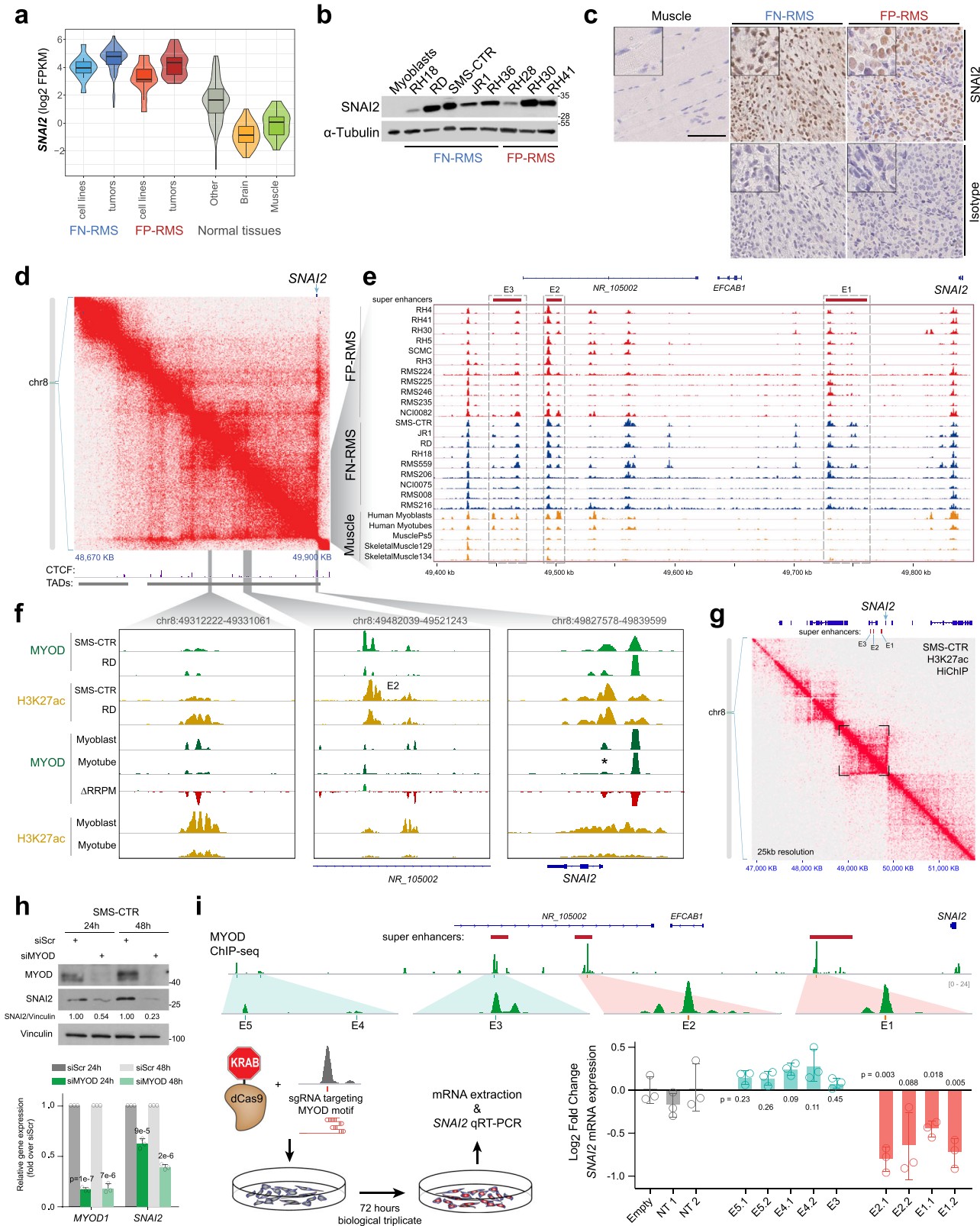

While FP-RMS and FN-RMS had similar enhancer structure surrounding *SNAI2* and transcriptomic levels of *SNAI2*, there is a selective dependency for *SNAI2* among FN-RMS cells when compared to a panel of cancer cell lines in genome-wide CRISPR screening data from the Achilles Project (Fig. S1g, h). Together our data provide evidence that *SNAI2* is expressed in RMS, is regulated by a 1.2 Mb regulatory locus to which muscle-specific

transcription factor MYOD can bind to maintain its expression and there is a dependency for *SNAI2* in FN-RMS cells.

**Suppression of *SNAI2* activates myogenic differentiation and suppresses stemness in vitro in FN-RMS.** To define the role of SNAI2 in FN-RMS differentiation, we abrogated SNAI2

**Fig. 1 SNAI2 is highly expressed in RMS and is regulated by a MYOD bound super enhancer. a** Violin plot showing expression (log$_2$FPKM) of *SNAI2* across RMS and normal tissue. FPKM, Fragments Per Kilo base of transcript per Million mapped reads. **b** Representative western blot (*n* = 3 biologically independent experiments) of SNAI2 expression in different RMS cell lines. **c** Representative SNAI2 immunohistochemical staining in RMS primary tumors (14 positive of 19 FN-RMS and 3 positive of 4 FP-RMS) compared to normal muscle and Isotype control antibody. Scale Bar = 100 μM. **d** Topological interactions (Hi-C data from IMR90)[17] of TAD containing the *SNAI2* locus. **e** H3K27ac ChIP-seq data at *SNAI2* TAD showing recurrent super enhancers (SEs) in a panel of FP-RMS (red), FN-RMS (blue) primary tumor samples and cell lines, myoblasts, myotubes, and skeletal muscle cells (yellow). Solid horizontal blocks show location of predicted super enhancers. **f** *SNAI2* promoter and enhancer bound by MYOD and loaded with active histone mark H3K27ac in FN-RMS cell lines. * Previously reported MYOD peak near *SNAI2*[21]. RRPM, Reference-adjusted Reads Per Million Mapped Reads. **g** Topological interactions in SMS-CTR (HiChIP of H3K27ac) of SEs in the *SNAI2* locus. **h** siRNA targeting *MYOD1* was used to knock down MYOD expression in SMS-CTR cells. MYOD and SNAI2 expression was detected by western blot (top) and qRT-PCR (bottom). Data was normalized to cells treated with scramble siRNA (*n* = 3 biologically independent experiments, data presented as mean values ± SD, Student's two-tailed *t*-test, exact *p* values are reported in the figure). **i** Targeted disruption of MYOD binding sites in enhancers surrounding the *SNAI2* gene in SMS-CTR cells, using sgRNAs to deliver dCas9-KRAB suppressor. Location of enhancers E1–E5 are shown above. Schematic of experimental workflow shown in bottom left. Bar chart of qRT-PCR measurements for SNAI2 expression after dCas9-KRAB perturbation with various guides is shown in the bottom right; *p* values shown were calculated among biological triplicates using *t*-test with Welch's correction. Error bars represent the SD among the 3 biologically independent replicates.

expression in FN-RMS using lentiviral vectors expressing short hairpin RNAs (shRNA)[25]. These resulted in 58–87% reduction of SNAI2 levels in shRNA-treated RD cells compared to scramble (Scr) (Fig. 2a). RD cells with stable SNAI2 knockdown (KD) when cultured in differentiation medium (DM; supplemented with 2% horse serum) for 3 days, showed a significant increase in differentiation, as assessed by morphology changes leading to elongated myofiber-like structures, de novo expression of the terminal muscle differentiation marker Myosin Heavy Chain 1 (MyHC) and up-regulation of MEF2C (Fig. 2b–d) compared to scramble shRNA controls. The percentage of MyHC positive cells were ~20-fold higher in shSNAI2.1 and ~8-fold higher in shSNAI2.2 cells compared to scramble shRNA cells (1.19% ± 0.41 for shScr vs 20.75% ± 2.39 for shSNAI2.1, and 11.33% ± 1.51 for shSNAI2.2) (Fig. 2d). Similar results were obtained in JR1 and SMS-CTR cells (Fig. S2a–d, i–l). Consistent with increased expression of MyHC-positive cells, real time qRT-PCR analysis in RD cells revealed that SNAI2 KD cells expressed reduced levels of *PAX7*, a marker of undifferentiated satellite muscle stem cells, and an increase in myogenic regulators *MYOD1*, *MYOG*, *MEF2C*, *CDKN1A* (p21), *MEF2D*, and *MHCb* (MyHC) (Fig. 2e and Fig. S2e). Importantly, while most myogenic transcription factors went up between 1.3 and 6-fold, differentiation marker MyHC expression increased by log folds in RD (Fig. 2e). Similar trends were observed in JR1 and SMS-CTR cells (Fig. S2e, m).

Notably, short-term SNAI2 KD in JR1, SMS-CTR, and RD18 cells maintained in growth medium (GM, supplemented with 10% serum) led to similar fold changes in the percentage of MyHC-positive cells compared to controls (shScr 0.56% ± 0.19, shSNAI2.1 6.41% ± 1.79, *p* = 0.04; shScr 0.24% ± 0.02, shSNAI2.1 2.10% ± 0.40, *p* = 0.02; and shScr 1.91% ± 0.28, shSNAI2.1 12.9% ± 1.57, *p* = 0.001, for JR1, RD18 and SMS-CTR cells, respectively; Student's two-tailed *t*-test) (Fig. S3b, c, e, f, h, i). This phenomenon was associated with the up-regulation of cell cycle arrest and differentiation markers at the protein and mRNA level including *CDKN1A, MYOG, MCK,* and *MyHC* (Fig. 2f and Fig. S3a, d, g, j–l). Next, we assessed if *SNAI2* is involved in stemness by determining sphere formation in FN-RMS[26]. Consistent with the induction of a myogenic differentiation, rhabdosphere formation was significantly reduced after knock-down of SNAI2 in RD, JR1, and SMS-CTR cells (shScr vs shSNAI2.1 average counts; 345.6 ± 19.7 vs 167.6 ± 3.5 spheres for RD, 3451 ± 122.05 vs 1337 ± 54.64 spheres for JR1, per 10,000 cells plated, and 41 ± 2.52 vs 12 ± 2.08 for SMS-CTR, per 20,000 cells plated) (Fig. 2g–i and Fig. S2 f–h, n–p).

Thus, *SNAI2* represses the expression of key myogenic regulators including *MYOD1, MYOG, MEF* family TFs, and *CDKN1A*, is a potent inhibitor of FN-RMS differentiation, and maintains stemness.

**Suppression of *SNAI2* reduces tumorigenicity and growth, and induces muscle differentiation in vitro and in vivo in FN-RMS, which is enhanced by vincristine.** To further define *SNAI2* function, we assessed the effect of its suppression on proliferation in vitro using proliferation assays, colony formation, and anchorage-independent growth in soft agar using shRNA against SNAI2 in FN-RMS RD, JR1, SMS-CTR, and RD18 cell lines. Transient SNAI2 KD in RD cells cultured in GM had significantly reduced proliferation compared to scramble shRNA control cells (Fig. 3a) (Change at day 10 for shScr 56.85 ± 3.72, shSNAI2.1 20.49 ± 1.84, and shSNAI2.2 14.95 ± 1.08,) and showed a 3-fold reduction in colony forming units (CFUs) (colonies/field: shScr 6.0 ± 1.0, shSNAI2.1 0.5 ± 0.57) and reduced soft agar colony forming ability (Fig. 3b–e). Similarly, short-term SNAI2-silenced JR1, RD18, and SMS-CTR cells cultured in GM also showed slower growth compared to scramble shRNA cells (Figs. S4a, f and S5a) and a reduction in soft agar anchorage-independent colony forming ability as well as in CFUs (Figs. S4b–e, g–j and S5b–e), indicating a consistent loss of proliferative as well as tumorigenic potential in FN-RMS upon SNAI2 depletion in transient knockdown assays.

We next assessed the effect of SNAI2 knockdown in vivo, employing RD, JR1, and SMS-CTR cells with scramble shRNA and SNAI2 shRNA xenografted subcutaneously in SCID mice. SNAI2 KD caused a significant reduction in the tumor size, weight and volume compared to control shScr. Indeed, by day 76 RD shSNAI2 tumors were significantly smaller in volume compared to Scr-transduced tumors in the same mice (Fig. 3f), and reduced size and weight compared to scramble shRNA tumors at the experimental end point (Fig. 3g–i) (volume shScr 1700.00 ± 585.42 vs shSNAI2.2 630.33 ± 139.75 mm$^3$; weight: shScr 2.7 ± 0.5 vs shSNAI2.2 1.29 ± 0.20 g). Similar results were obtained for JR1 (shScr 1439 ± 295 vs shSNAI2.1 707 ± 145 mm$^3$) and SMS-CTR xenografts (shScr 177.1 ± 127 vs shSNAI2.2 17.7 ± 18.96 mm$^3$) (Figs. S4k–m and S5f–i).

Hematoxylin and eosin (H&E) staining of the tumors revealed that the SNAI2 knockdown tumors were less dense (Fig. 3j, k and Figs. S4n, o and S5j, k) and more differentiated, as evidenced by increased MyHC staining (percentage of MyHC-stained cells/field, shScr 3.7 ± 2.4 vs shSNAI2.2 10.7 ± 3.9, *p* = 0.0003; Student's two-tailed *t*-test) (Fig. 3l, m and Figs. S4p, q and S5l, m). Altogether, these results strongly support a role for *SNAI2* on tumor growth and as a potent suppressor of myogenic differentiation in FN-RMS.

Since SNAI2 KD cells express higher levels of myogenic differentiation transcription factors MYOD, MYOG, and MEF2C, we hypothesized that combining *SNAI2* silencing with a standard of care drug in RMS therapy would enhance differentiation of

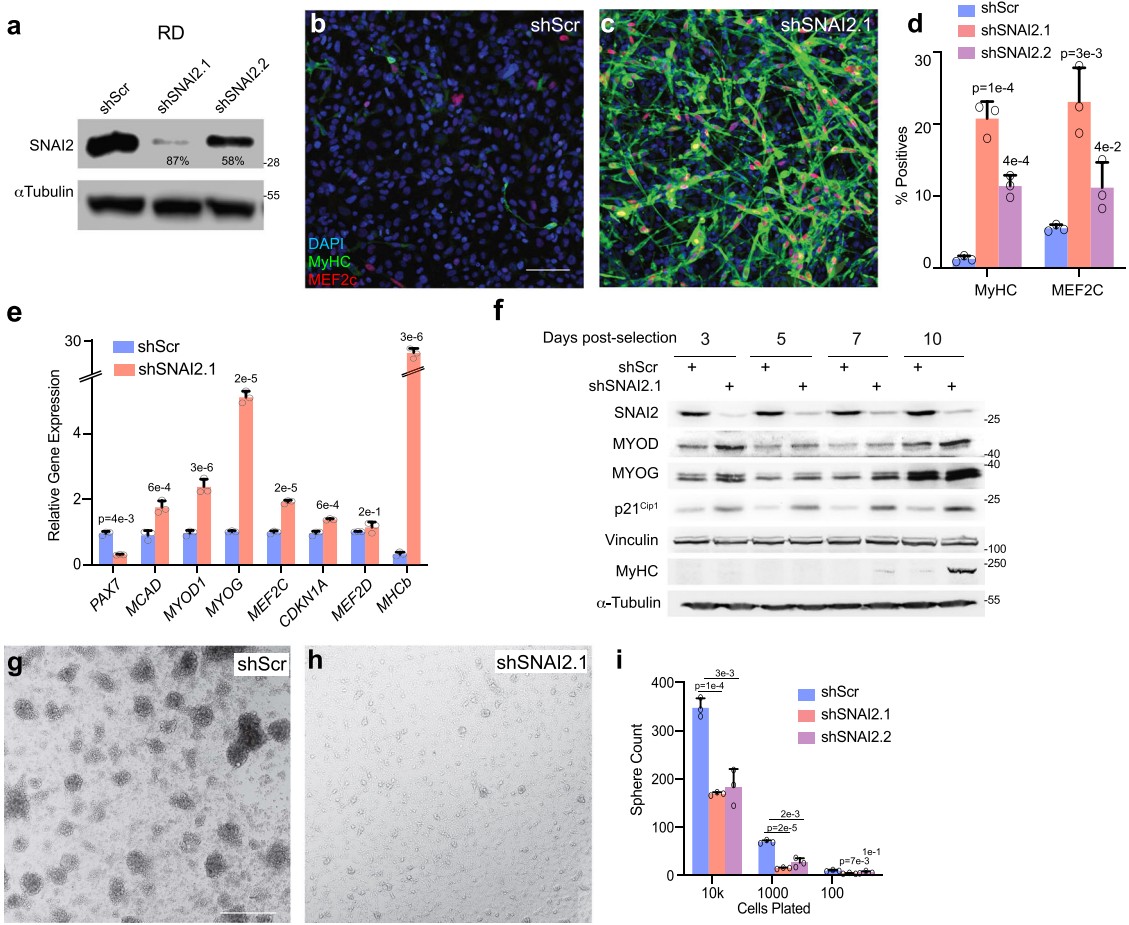

**Fig. 2 Suppression of *SNAI2* activates myogenic differentiation and suppresses stemness in vitro in FN-RMS. a** The level of SNAI2 knockdown by shSNAI2 in RD cells compared to shScr assessed by western blot (Representative blot, $n = 3$ biologically independent experiments). **b** Representative images of immunostaining in RD cells stably expressing shScr or **c** shSNAI2.1 shRNA stained for Myosin Heavy Chain 1 (MyHC, green), MEF2C (red) and DAPI for nuclei (blue). **d** Quantitation of immunostaining counts as percentage values to total nuclei per image ($n = 3$ biologically independent experiments, data presented as mean values ± SD, Student's two-tailed *t*-test, exact *p* values are reported in the figure). **e** qRT-PCR gene expression analysis in RD cells comparing shScr to shSNAI2.1 KD showing early and late myogenic markers ($n = 3$ biologically independent experiments, data presented as mean values ± SD, Student's two-tailed *t*-test, exact *p* values are reported in the figure). **f** Representative western blot ($n = 3$ biologically independent experiments) of muscle differentiation genes in RD shScr vs shSNAI2.1 transient KD cells at 3, 5, 7, and 10 days post puromycin selection. **g, h** Representative images of sphere formation assays in RD cells containing shScr or shSNAI2.1. **i** Quantitation of sphere counts in RD cells plated at three densities (10,000, 1000, and 100 per well). $n = 3$ biologically independent experiments, data presented as mean values ± SD, Student's two-tailed *t*-test, exact *p* values are reported in the figure, Scale Bars, b, g = 100 μM.

tumor cells often detected as nests of cells after conventional therapy[27]. Therefore, we performed xenograft transplants of Scr and SNAI2 shRNA knockdown cells into SCID mice and once tumors were palpable, mice were treated with a subtherapeutic dose of vincristine (0.5 mg/kg) once a week for 3 weeks[28] and the regrowth of tumor after the end of drug treatment was assessed. We observed that while tumors from both Scr and SNAI2 shRNA treated cells initially decreased in volume, shScr tumors very quickly regrew (Fig. 3n–q and Fig. S4r, s). However, SNAI2 knockdown tumors were unable to grow back quickly and were significantly reduced in volume (Fig. 3n–q and Fig. S4r, s). This reduction in volume was much more significant than in untreated xenograft experiments. To assess the effect of vincristine on differentiation, a subset of shScr and SNAI2 shRNA-tumor burdened mice were given a single dose of the drug and animals were euthanized and tumors harvested 24 h later and processed for H&E and IHC staining for differentiated myosin (MyHC). H&E staining on SNAI2 knockdown tumors showed features associated with differentiating myoblasts when compared to

scramble treated tumors indicating an increase in differentiation (Fig. 3r, s). The effect on MyHC expression was more extreme in shSNAI2.2 cells, which had ~15-fold increase in differentiation than control-treated tumors (Fig. 3t, u) (shScr vs shSNAI2.2 2.8 ± 0.8% vs 41.0 ± 2.9% of MyHC stained cells/field, $p = 0.00003$; Student's two-tailed *t*-test). These findings support the evidence that *SNAI2* is a potent inhibitor of differentiation in vivo. Moreover, SNAI2 knockdown cells are poised towards differentiation, and in this context, treatment with vincristine can synergistically enhance differentiation and exit of tumor cells from the cell cycle resulting in significant reduction in tumor growth.

**SNAI2 binds key enhancers in FN-RMS.** Our evidence that SNAI2 KD induces differentiation in FN-RMS, combined with the knowledge that Snail proteins can bind to E-box motifs and are rapidly switched-off during normal muscle differentiation[29], led us to hypothesize that SNAI2, by binding to key sites across

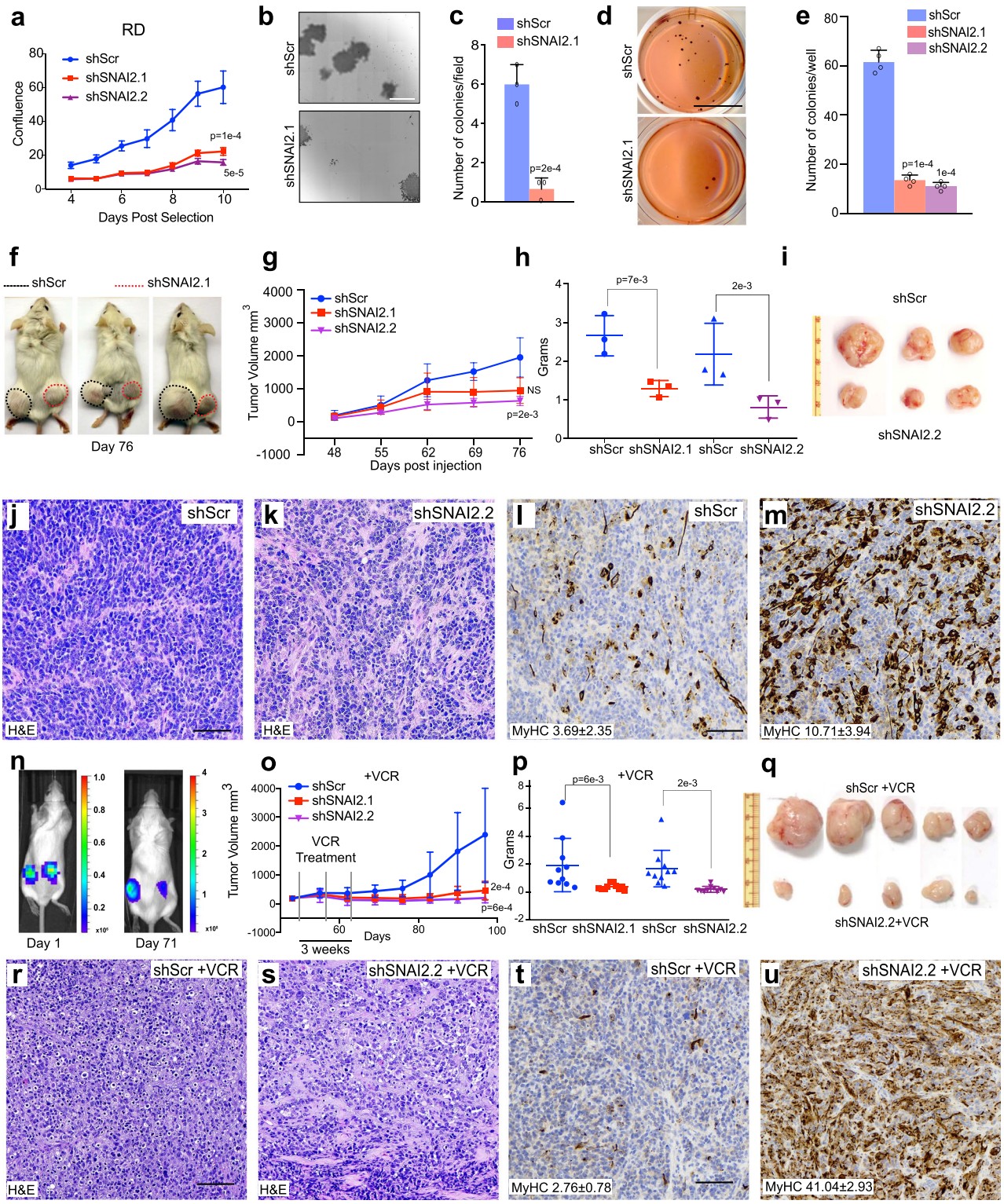

the RMS epigenome, arrests myogenic differentiation. To explore this, we mapped the genome-wide profile of SNAI2 chromatin binding, using chromatin immunoprecipitation followed by sequencing (ChIP-seq). To identify high-confidence SNAI2 sites, we performed ChIP-seq in three FN-RMS cell lines, SMS-CTR, RD and JR1. We identified 1069 peaks shared among at least 2 cell lines that included 146 peaks common to all the 3 cell lines (Fig. S6a), and these showed the strongest signal per peak, as exemplified on the regulatory regions of the myogenic gene *RYR1*

(Fig. 4a). SNAI2-bound loci contained the SNAI2 DNA binding motif ($p = $1e-700, HOMER motif analysis)[30]; however, SNAI2 was also enriched in E-box motifs for known basic helix-loop-helix (bHLH) myogenic transcription factors such as MYOD, MYOG, and MYF5 (Fig. 4b), indicating potential co-binding or adjacent binding to myogenic TFs.

To determine the types of chromatin occupied by SNAI2, we used patterns of histone marks and CTCF in SMS-CTR cells[19] to define chromatin states with ChromHMM[31]. SNAI2 associated

**Fig. 3 Suppression of *SNAI2* reduces tumorigenicity and growth, and induces muscle differentiation in vitro and in vivo in FN-RMS. a** Growth curve analysis of RD cells 3 days post puromycin selection after lentiviral infection with shScr or shSNAI2 shRNAs ($n = 3$ biologically independent experiments, data presented as mean values ± SD, Student's two-tailed $t$-test, exact $p$ values are reported in the figure). **b, c** Single cell colony formation assay in RD cells containing shScr or shSNAI2 knockdown and quantitation values of colony forming units ($n = 3$ biologically independent experiments, data presented as mean values ± SD, Student's two-tailed $t$-test, exact $p$ values are reported in the figure). **d, e** Soft agar colony formation assay comparing RD shScr to shSNAI2 infected cells and quantification of colony numbers in wells ($n = 3$ biologically independent experiments, data presented as mean values ± SD, Student's two-tailed $t$-test, exact $p$ values are reported in the figure). **f** RD cells xenografted subcutaneously in mice with shScr (left) or shSNAI2 (right) and followed for 76 days (3 representative mice of 6 shScr, 3 shSNAI2.1, and 3 shSNAI2.2 tumors each, $1 \times 10^6$ cells). **g** Tumor volume of mice injected with either shScr or shSNAI2 cells assessed weekly by caliper measurement represented as mm$^3$ ($n = 6$ shScr, $n = 3$ shSNAI2.1, and $n = 3$ shSNAI2.2 biologically independent experiments, data presented as mean values ± SD, Student's two-tailed $t$-test, exact $p$ values are reported in the figure). **h** Weight measurement of xenograft tumors with either shScr or shSNAI2.1, SNAI2.2 post mortem ($n = 3$ biologically independent experiments, data presented as mean values ± SD, Student's two-tailed $t$-test, exact $p$ values are reported in the figure). **i** Images of shScr and shSNAI2.2 RD tumors taken at 76 days. **j–m** Hematoxylin and eosin and Immunohistochemistry of MyHC in the same tumors (Representative images of $n = 3$ biologically independent experiments.) **n** Growth of vincristine treated (0.5 mg/kg once weekly for 3 weeks) RD tumors expressing shScr or shSNAI2 assessed by Luciferase imaging. **o** Tumor volume of transplanted RD xenografts with shScr and shSNAI2 + vincristine (VCR) assessed by caliper measurement represented in mm$^3$ followed for 96 days ($n = 10$ biologically independent experiments, data presented as mean values ± SD, Student's two-tailed $t$-test, exact $p$ values are reported in the figure). **p** Tumor weight of RD xenografts expressing shScr or shSNAI2 treated with VCR and harvested at 96 days ($n = 20$ mice, 20 shScr, 10 shSNAI2.1 and 10 shSNAI2.2 tumors from biologically independent experiments, data presented as mean values ± SD, Student's two-tailed $t$-test, exact $p$ values are reported in the figure). **q** Images of shScr and shSNAI2.2 tumors extracted from mice post euthanasia. **r–u** H&E and MyHC immunohistochemistry of tumor sections from RD xenografts expressing shScr or shSNAI2 treated with vincristine (VCR). Representative images of $n = 3$ biologically independent experiments. Scale Bars in j, l, r, t = 100 μM, Scale bar in b = 100 μM, d = 10 mm.

primarily with the strong enhancer chromatin state in FN-RMS, characterized by H3K27ac and H3K4me1/2 histone modifications, and secondarily to poised promoters (Fig. 4c). Because SNAI2 binds to E-box motifs that share similar chromatin localization pattern with MYOD[20], we investigated the overlap of SNAI2 and MYOD genome-wide binding in SMS-CTR, RD, and JR1. We defined unique SNAI2 (405, blue), unique MYOD (6689, green), and SNAI2-MYOD overlapping (664, olive) peaks (Fig. 4d). The shared peaks showed the highest SNAI2 and MYOD average signals compared to SNAI2 and MYOD unique peaks (Fig. 4d). Motif analysis of the SNAI2 and MYOD unique (only) and the overlapping peaks revealed highly enriched motifs for known myogenic transcription factors including MYF5 and MYOG in all three groups (Fig. S6b). We found low abundance of the SNAI2 motif in MYOD-only peaks as well as low enrichment of the MYOD motif in SNAI2-only peaks (Fig. S6b). Next, based on chromatin-state association profile, we mapped SNAI2 and MYOD occupancy at enhancer sites. Using the ROSE2 (Rank Order of Super Enhancers) algorithm, we classified the enhancers into typical ($n = 8223$) and super ($n = 516$, SEs) (Fig. S6c). SNAI2 localizes at SE regions only at MYOD peaks, owing to the ubiquitous presence of MYOD in SEs, while only a few typical enhancers had SNAI2 alone.

Functional annotation of the MYOD and SNAI2 targets[32] revealed few MYOD-specific or SNAI2-specific processes, while most are common and indicate a strong association with muscle-specific processes (Fig. S6d). A large category of genes are involved in MAP kinase activity cascade, a pathway in RAS-driven FN-RMS known to block myogenesis[20]. Thus, SNAI2 preferentially binds chromatin associated with enhancers through a repertoire of E-box motifs including the known SNAI2 motif, and the majority of SNAI2 binding is shared with MYOD.

In order to define whether MYOD and SNAI2 co-occupy chromatin spatially, as evidenced by ChIP-seq, and also co-temporally, we performed tandem chromatin immunoprecipitation (re-ChIP) for MYOD followed by SNAI2, and vice versa, at *MYOG* associated promoter and super enhancers in SMS-CTR cells. Re-ChIP qPCR signal at regions bound by both MYOD and SNAI2 (*MYOG* SE1 and SE2) revealed strong enrichment irrespective of ChIP order (Fig. 4e). This signal is absent at MYOD-only (*MYOG* promoter) and SNAI2-only (*MRPL9*) binding sites. Thus, MYOD and SNAI2 are co-present at key

regulatory chromatin locations of *MYOG* at the same time regulating its expression, possibly through forming a transcriptional condensate or a high-order complex at these regulatory elements.

**Ablation of SNAI2 enables MYOD to activate myogenic target genes.** Given that SNAI2 binding overlaps with super enhancers and enhancers through E-boxes associated with MYOD, we hypothesized that SNAI2 could be hindering MYOD from fully binding to key myogenic enhancers, thus preventing differentiation in FN-RMS. To investigate this, we performed ChIP-seq for SNAI2 and MYOD in control (scramble shRNA) and SNAI2 knockdown conditions in SMS-CTR and RD cells. shRNA knockdown resulted in reduction of SNAI2 at its cognate sites (Fig. 5a, b). Genome-wide assessment of MYOD occupancy revealed that MYOD deposition did not change overall at either promoters, regular enhancers or SEs (Fig. S7a). However, we found that MYOD deposition specifically increased upon SNAI2 knockdown at peaks overlapping SNAI2 (Fig. 5c and Fig. S7a, b). We therefore sought to define the identity and consequences on expression of the subset of genes regulated by E-boxes at which SNAI2 competes with MYOD for binding.

Previously, assessing the expression of critical myogenic regulators including *MYOD1, MYOG, MCAD, MEF2C, MEF2D*, and *CDKN1A* (Fig. 2e) we demonstrated that SNAI2 blocks expression of genes essential for myogenic differentiation. To more comprehensively and in an unbiased manner connect chromatin-binding events with all possible SNAI2-driven transcriptional changes, we next performed RNA-seq together with gene set enrichment analysis (GSEA) upon SNAI2 knockdown in SMS-CTR and RD cells. The most significantly enriched pathways (FDR $q = 0$) were Hallmark myogenesis genes, including genes up-regulated during in vitro differentiation from myoblast to myotubes and genes driven by MYOG-activated SEs during FN-RMS treatment with trametinib[20] (Fig. 5d). Conversely, the analysis showed negative enrichment for genes down-regulated during myoblast differentiation (Fig. 5d and Fig. S7c).

While many myogenic genes are up-regulated after SNAI2 knockdown, these likely represent a mixture of genes both directly repressed by SNAI2 and others indirectly activated, perhaps downstream of a direct SNAI2 target that becomes active. To clarify this, we integrated genomic binding data with

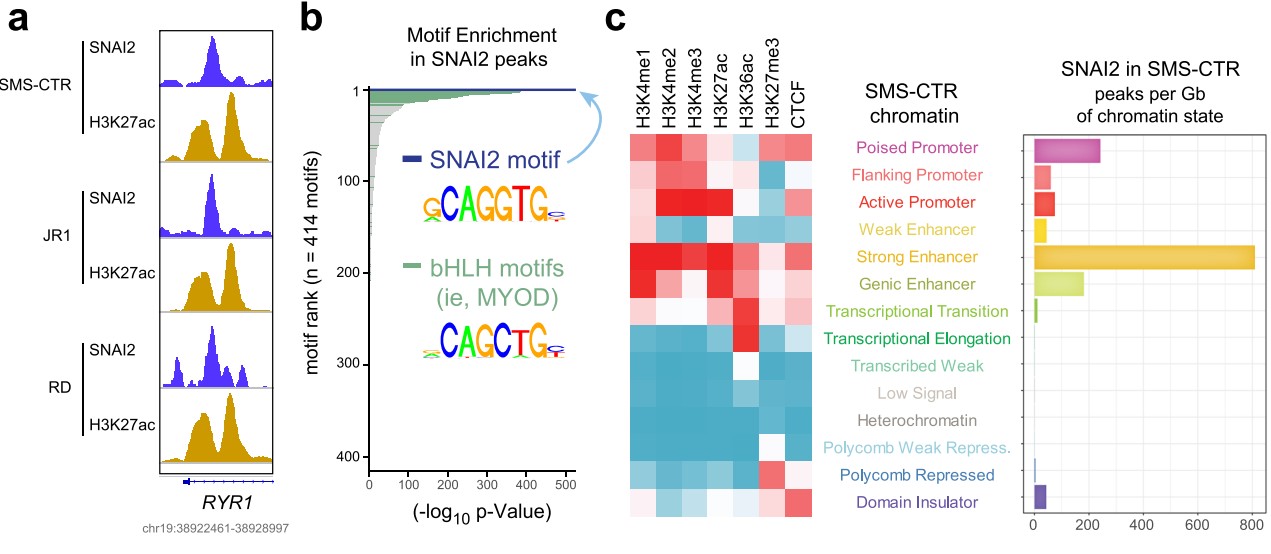

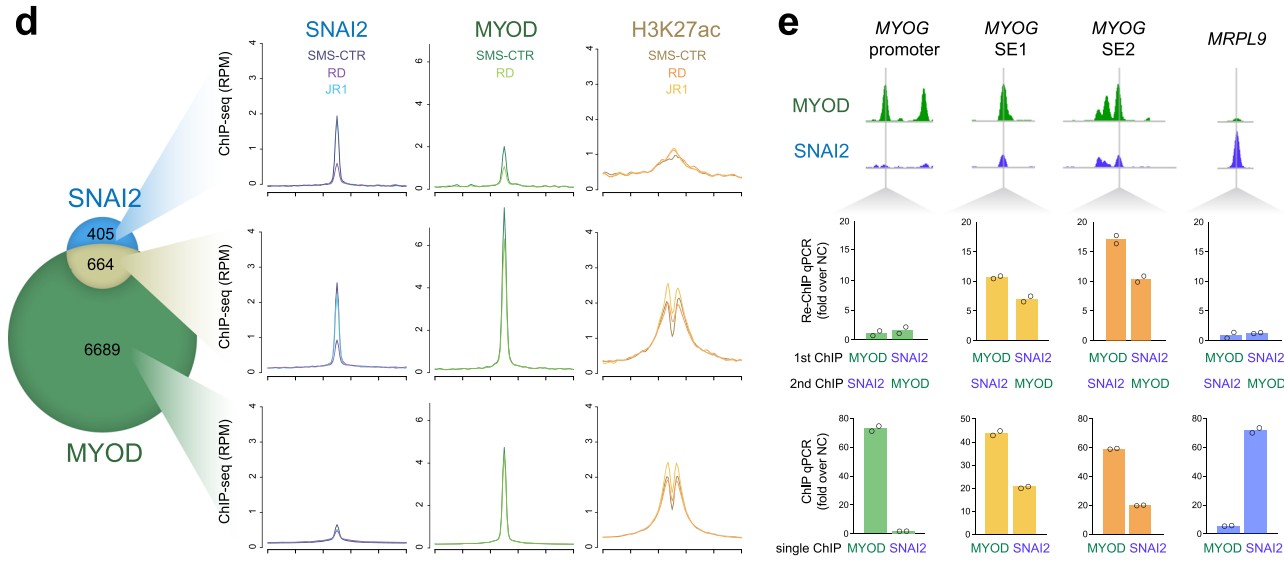

**Fig. 4 SNAI2 binds key enhancers in FN-RMS. a** ChIP-seq signal for SNAI2 and H3K27ac peaks shared in all three cell lines are shown. **b** Hypergeometric Optimization of Motif EnRichment (HOMER) analysis identified SNAI2 binding motifs (top) as well as bHLH (basic helix-loop-helix) motifs at SNAI2 shared peaks using the HOMER package (homer.salk.edu/homer/ngs/peakMotifs.html). *p* statistic is calculated using the HOMER statistical comparison against size matched DNA sequences from randomly selected background genomic sequences. **c** Chromatin states in SMS-CTR cells (left) and abundance of SNAI2 peaks per Gb of each state (right). **d** Venn diagram (left) and average plot for ChIP-seq signal (right) depicting overlap between SNAI2 (2 or all 3 cell lines) and MYOD (2 cell lines) binding sites. RPM, Reads Per Million Mapped Reads. **e** Co-occupancy of MYOD and SNAI2 as measured by ChIP-reChIP and qPCR at locations previously identified (by ChIP-seq) as being preferentially bound by MYOD, SNAI2, or both. Single-target ChIP-qPCR controls are shown below (*n* = 2 biologically independent experiments, data presented as mean values).

transcriptional data. We matched high-confidence SNAI2 peaks in FN-RMS cell lines (*n* = 1069, see Fig. S6a) with all candidate target genes in the RNA-seq data, constrained to genes within each SNAI2-containing TAD (predicted by Hi-C data[17]). We considered myogenic genes that were consistently up-regulated in both RD and SMS-CTR (*n* = 128). We found 29 high-confidence SNAI2 myogenic directly suppressed genes (Fig. 5e and Supplementary Table 1) including *MYOG*, *MEF2A*, and *RASSF4* as well as indirect targets (*n* = 99) that lack a SNAI2 peak anywhere within their TAD. Indirect targets include muscle differentiation transcription factors *MEF2C*, *CDKN1A*, and *RB1*. The positive regulation of most of these genes has been already associated with a terminal differentiation program in muscle and

in RMS, and are activated directly by MYOG via super enhancers[20,29,33], suggesting MYOG (a direct SNAI2 target) mediates the activation of these indirect targets once up-regulated by SNAI2 knockdown. Interestingly, SNAI2 binds to myogenic enhancers with MYOD at *MYOG*, *MYBPH*, *TNNT1/2*, *MAPKAPK2*, and *MEF2A*, which are all activated during shSNAI2-induced differentiation in FN-RMS (Fig. 5f and Fig. S7d). We looked at *MYOG* and *MEF2A* and found they each have enhancers that are proximal in 3D (assessed by H3K27ac HiChIP in SMS-CTR cells, Fig. 5f) that are bound by SNAI2. Once SNAI2 is diminished upon knockdown, focused growth of MYOD binding occurs at these sites of SNAI2 loss (Fig. 5f) and the expression of these genes are robustly activated (the *MYOG*

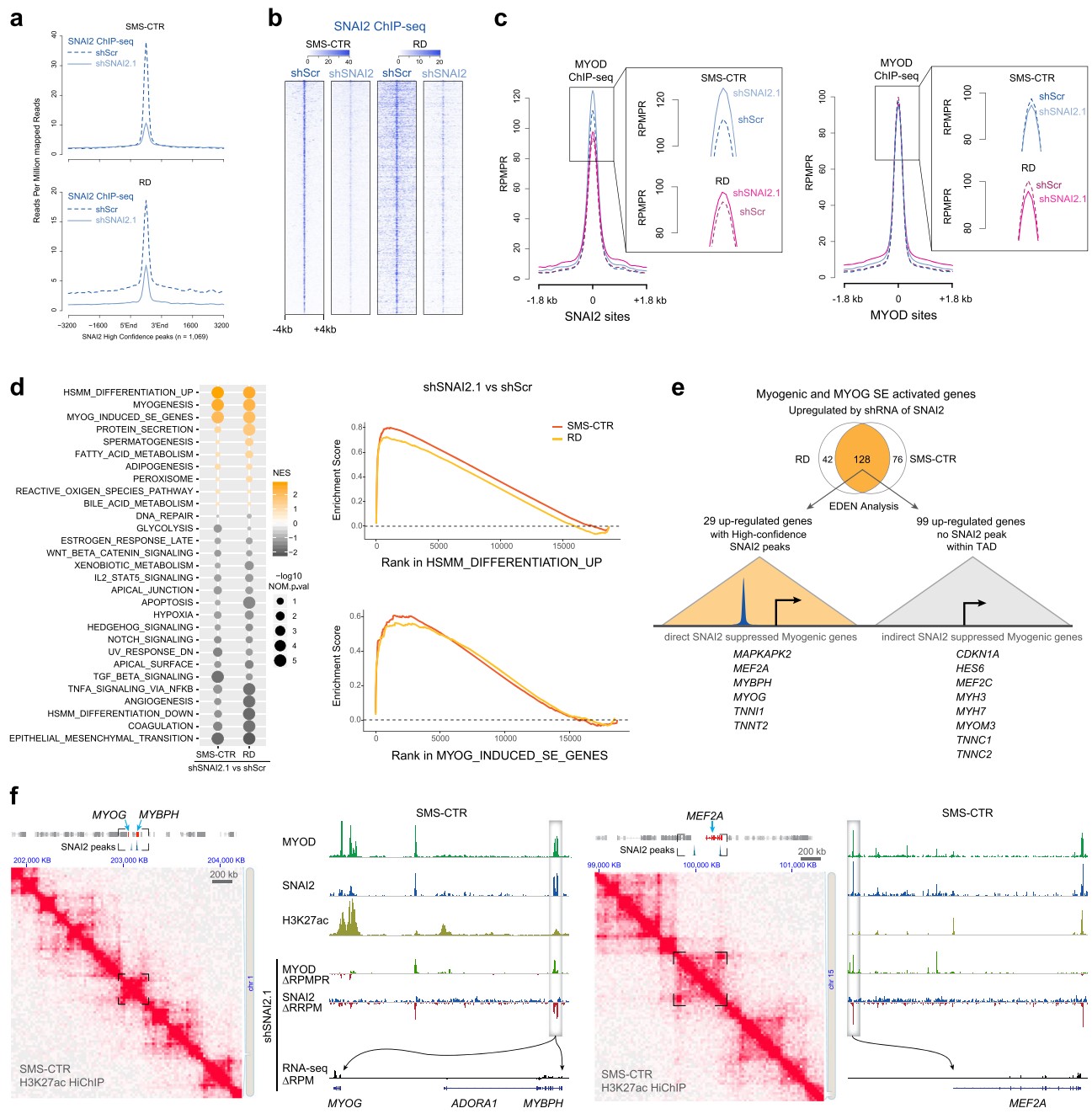

**Fig. 5 Ablation of SNAI2 enables MYOD to activate myogenic target genes. a** Composite plots showing SNAI2 signal intensities (reads per million mapped reads) at SNAI2 high confidence peaks (*n* = 1069) in SMS-CTR (top) and RD (bottom). **b** Heatmaps of SNAI2 peak intensity at SNAI2 high confidence peaks. Each row represents a genomic location and is centered around SNAI2 peaks, extended 4 kb in each direction, and sorted by SNAI2 signal strength. **c** Composite plots showing MYOD signal intensities at SNAI2 high confidence peaks (left) and at MYOD peaks (right) in SMS-CTR and RD. RPMPR, Reads Per Million Mapped Peak Reads. **d** Bubble plot depicting Gene Set Enrichment Analysis (GSEA) in SMS-CTR and RD cells (left). The size of the bubble is proportional to the −log10 nominal (NOM) *p*, and the color of the bubble corresponds to the normalized enrichment score (NES) value. GSEA enrichment plots showing positive enrichment for a set of genes up-regulated during differentiation of human skeletal muscle myoblasts into myotubes (right, top), and a set of myogenically induced super-enhancer genes (right, bottom). *p* values are determined by the GSEA algorithm relative to the null distribution calculated with 1000 permutations. For each of the enrichment plots shown here, the false discovery rate (FDR) *q* value and the nominal *p* value is <0.0005. **e** Diagram illustrating SNAI2 direct or indirect myogenic target genes through EDEN analysis. SE, Super Enhancer; TAD, Topologically Associated Domain. **f** Sites of direct SNAI2 mediated gene suppression at *MYOG*, *MYBPH* (left), and *MEF2A* (right), with both H3K27ac HiChIP for 3D chromatin folding and ChIP-seq. Representative ChIP-seq tracks are shown for MYOD, SNAI2, H3K27ac, and delta (Δ) value (shSNAI2.1 minus shScr) in MYOD and SNAI2 and gene expression (RNA-seq) at *MYOG*, *MYBPH*, and *MEF2A* loci in SMS-CTR. Arrows depict SNAI2/MYOD regulation on direct target myogenic genes. RPMPR, Reads Per Million Mapped Peak Reads; RRPM, Reference-adjusted Reads Per Million Mapped Reads; RPM, Reads Per Million Mapped Reads.

enhancer for example, Fig. 5f). Taken together, our data indicate that SNAI2 through competition with MYOD directly blocks terminal differentiation. Additionally, through indirect mechanisms, SNAI2 also inhibits the expression of *MEF2C* and *CDKN1A* that are normally repressed in FN-RMS.

**SNAI2 anti-differentiation effects are mediated by the blockade of *MYOG*, *MEF2A/C/D*, and *CDKN1A* gene expression.** Our data suggest that *SNAI2* supports FN-RMS growth and blocks differentiation (Fig. 6a) through direct repression of master TFs of myogenesis *MYOG* and *MEF2A*, and indirect suppression of *CDKN1A* and *MEF2C/D*. To test this model, we determined the effects of inhibiting expression of *MYOG* and *MEF2A* (SNAI2 direct target genes) as well as *MEF2D* and *CDKN1A* (SNAI2 indirect target genes) on blocking differentiation in SNAI2 knockdown cells. We performed siRNA-mediated gene-specific knockdown in SNAI2 shRNA-silenced cells and used scramble shRNA cells as controls. The siRNA-transfected cells were switched to differentiation medium 24 h post-transfection and differentiation was assessed 72 h later. Compared to control scramble siRNA-transfected cells, siRNA knockdown of *MYOG*, *MEF2A*, *MEF2D*, and *CDKN1A* in shSNAI2 setting (Fig. 6d, g and Fig. S8a, b, i, j) blocked the expression of differentiated myosin MyHC in both RD (MyHC positive cells: siScr 13.2 ± 1.1%, siMEF2A 2.1 ± 1.5%, siMYOG 1.8 ± 0.9%, siCDKN1A 7.6 ± 0.7%, siMEF2D 2.3 ± 0.2% with *p* values: MYOG (0.0002), MEF2A (0.0005), MEF2D (0.0003), and CDKN1A (0.006) siRNA, Student's two-tailed *t*-test) (Fig. 6b, c, e, f, h, i and Fig. S8c–e, f–h) and JR1 cells (MyHC positive cells: siScr 10.2 ± 0.5%, siMEF2A 1.0 ± 0.2%, siMYOG 0.8 ± 0.2%, siCDKN1A 0.7 ± 0.6%, siMEF2D 1.1 ± 0.5%) (Fig. S8k–m, n–q). Of the four genes assessed, *MYOG*, *MEF2D*, and *MEF2A* silencing had the biggest effect on suppressing differentiation, while the effect of *CDKN1A* on differentiation although significant was less robust. Thus, SNAI2 blocks differentiation of FN-RMS cells through direct down-regulation and suppression of *MYOG* and *MEF2A* activity and indirectly by repressing *MEF2D* and *CDKN1A*.

MEK inhibitor trametinib induces differentiation in RAS-mutated FN-RMS through unlocking the expression of pro-differentiation myogenic transcription factors[20]. Due to the importance of RAS as an oncogenic driver in FN-RMS, we sought to evaluate if the effects of SNAI2 on differentiation, in the same tumor context, involved the RAF-MEK-ERK pathway. We determined the expression of myogenic genes identified here as direct SNAI2 targets and found that these genes are also up-regulated upon trametinib treatment (shSNAI2 vs shScr: *p* = 0.0001, trametinib vs DMSO: *p* = 0.0001; Student's two-tailed *t*-test) (Fig. 6j).

RAS signal inhibition causes both myogenic SE genes to be activated via MYOG upregulation, but also shuts down RAS-dependent SEs at negative signal regulation genes[20]. We found that for MYOG-activated SE genes[20] that were altered by knockdown of SNAI2, their expression strongly correlated with those altered by chemical inhibition of the RAS pathway by the MEK inhibitor trametinib (Fig. 6k upper, for MYOG-activated SE genes: Pearson 0.636, *p* ≤ 0.0001). A similar phenomenon was seen for RAS-dependent SE genes albeit with less significance[20] (Fig. 6k lower, for RAS-dependent SE genes: Pearson 0.1786, *p* = 0.0087). Our data indicate that SNAI2 KD phenocopies trametinib-induced differentiation.

We next asked, could trametinib be achieving its up-regulation of myogenic targets through SNAI2 function impairment? To address this, we performed ChIP-seq of SNAI2 after trametinib treatment and focused our attention on myogenic enhancers bound by SNAI2 (reported in Fig. 5f). ChIP-seq profiles showed that SNAI2 binding to chromatin was reduced by the MEKi,

corresponding to trametinib-induced increase in MYOD binding, H3K27ac abundance, and expression of *MYOG*, *MYBPH*, *TNNT1*, and *MEF2A* (Fig. 6l). These effects are all seen during shSNAI2-induced differentiation in FN-RMS (Fig. 6m). Moreover, treatment of FN-RMS RD, JR1, and SMS-CTR cells with trametinib results in downregulation of SNAI2 expression by 72 h associated with a loss of phosphorylated ERK1/2 with no effects on total ERK expression (Fig. 6n). Similar to SNAI2-silenced cells, trametinib treated cells also show increased expression of MEF2C (Fig. 6n). Collectively, these results led us to a model where either MEK inhibition or shSNAI2 phenocopy one another through activation of *MYOG* and a *MEF2*-program that induces myogenic differentiation; this convergence is evidenced by the prevention of muscle differentiation by shSNAI2 by silencing of *MYOG* or *MEF2A* (Fig. 6o).

## Discussion

In this study, we report that SNAI2 acts as a bona fide oncogene in FN-RMS with several lines of evidence. Firstly, *SNAI2* is highly expressed in FN-RMS tumors compared to normal muscle and the majority of other cancers. Secondly, FN-RMS cells show the most dependency for SNAI2 compared to all other tumor cell lines, including FP-RMS, in the Broad Achilles CRISPR/Cas9 screen for core dependencies. Additionally, we find that knockdown of SNAI2 results in a robust induction of the myogenic master transcription factors *MYOD1* and *MYOG* with increase in structural muscle genes and in the potent cyclin-dependent kinase inhibitor *CDKN1A*. This is accompanied by reduced stemness, tumorigenicity, and growth together with evidence of muscle differentiation both in vivo and in vitro.

We dissect SNAI2-dependent mechanisms in the context of FN-RMS and the muscle differentiation pathway. We report that *SNAI2* acts through a gene regulatory network in which its expression is modulated by muscle-lineage master transcription factor MYOD and in turn SNAI2 regulates both MYOD-dependent and MYOD-independent programs governing cell differentiation and growth.

We find that *SNAI2* expression is modulated by a large 1.2 Mb regulatory TAD region, where binding of the enhancer mark H3K27ac is correlated with expression of *SNAI2*. In development, normal myoblasts show high enhancer activity in this region associated with high expression of *SNAI2*, which is switched-off in differentiating myotubes and in terminally differentiated muscle. Further, while many pathways and genes are known to regulate SNAI2 in different cell types and tumors[34], in FN-RMS tumors we find that MYOD can also bind to *SNAI2* enhancers and can directly up-regulate its expression. This is especially interesting as it has been previously shown that FN-RMS tumors robustly express MYOD and are dependent on MYOD function for tumorigenesis[35].

A critical finding of our study is that SNAI2 binding to chromatin is directly linked to differential gene expression and consequently cellular function. Combining ChIP-seq and RNA-seq datasets for SNAI2 and MYOD finds that SNAI2 binds a repertoire of E-box DNA elements, including the SNAI2 E-box and E-box motifs to which MYOD, MYOG, and MYF5 are known to bind. Importantly, SNAI2 engaged *cis*-regulatory elements are specifically associated with super enhancers. In the context of FN-RMS, many of the enhancers that SNAI2 engages are associated with myogenic differentiation genes where the binding of SNAI2 competes with that of MYOD. Thus, the repressor function of SNAI2, through selective competition with transcriptional activator function of MYOD, results in enhancer dampening at loci regulating genes driving early muscle differentiation (for example, *MYOG* and *MEF2A*) and terminal differentiation genes (for

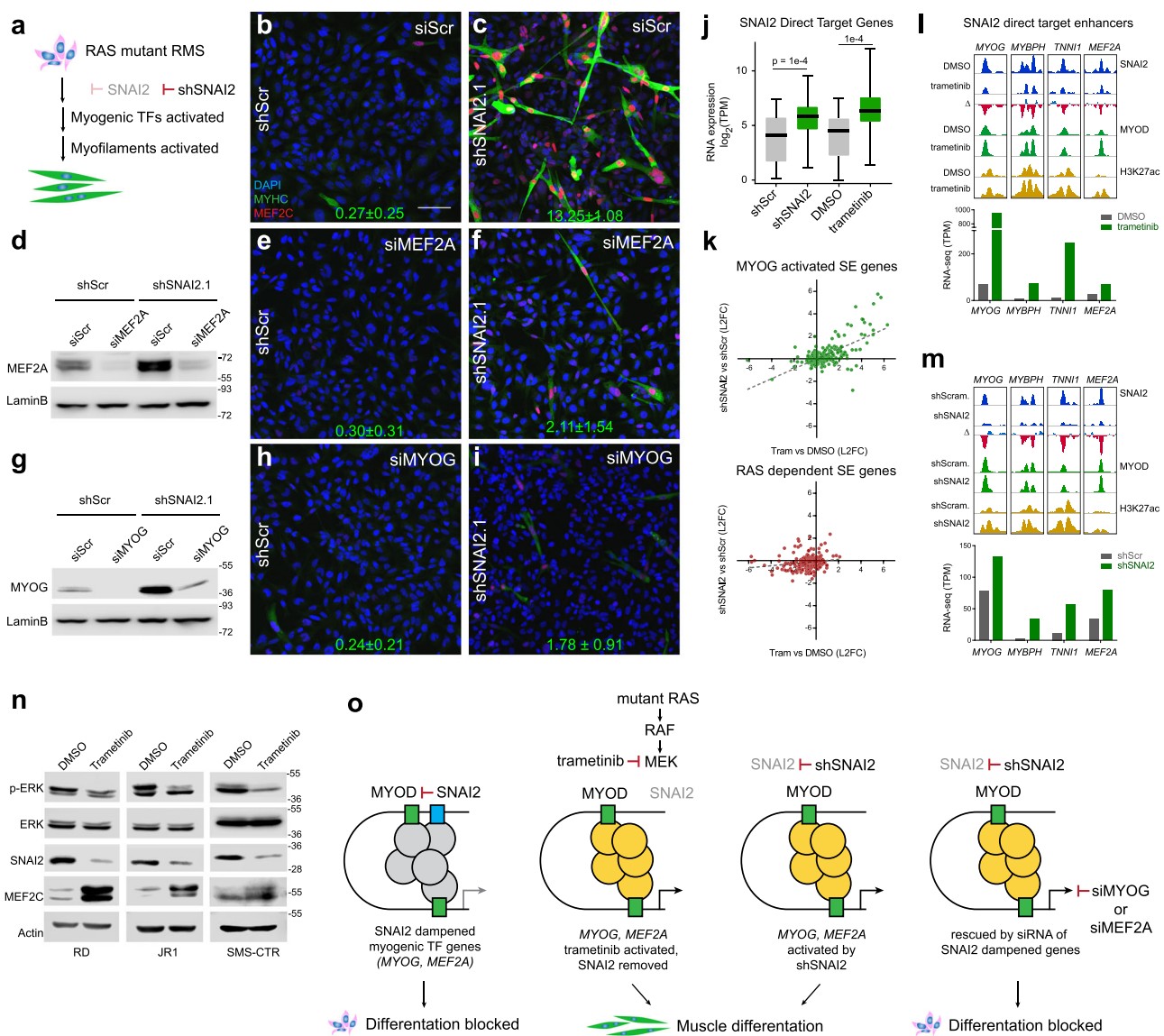

**Fig. 6 *SNAI2* anti-differentiation effects are mediated by the blockade of *MYOG, MEF2A/C/D* and *CDKN1A*. a** Schematic representation of role of *SNAI2* in RAS-mutated RMS and induction of differentiation after SNAI2 knock down. **b, c, e, f, h, i** Representative images of RD shScr and shSNAI2 cells transfected with siRNA stained for Myosin Heavy Chain 1 (MyHC, green), MEF2C (red) and DAPI for nuclei (blue). Green numbers bottom of each image represents the average number of percentage of positive cells for three images. Scale Bar in b = 100 μM. **d, g** Western blot of RD shScr and shSNAI2.1 cells transfected with siRNAs against MEF2A, and MYOG along with Control siRNA (scramble) probed for MEF2A and MYOG. Representative blot, n = 3 biologically independent experiments. **j** Box plot depicting SNAI2 myogenic direct target gene expressions (RNA-seq) in SMS-CTR cells transduced with shRNAs or treated with trametinib. TPM, Transcripts per Million. Box plots show quartiles, black bar shows the median, and whiskers show the 1.5 × interquartile range. The exact *p* values are reported in the figure (n = 3 biologically independent experiments). **k** Scatter plot of log2 fold change (L2FC) of MYOG-activated SE gene expression (RNA-seq)[20] (top) and shSNAI2/shScr transduced vs trametinib/DMSO treated cells for RAS-dependent SE gene expression (RNA-seq)[20] (bottom) in SMS-CTR cells. **l** Representative ChIP-seq tracks for SNAI2 (blue), MYOD (green), H3K27ac (yellow) (top) and gene expression (RNA-seq) (bottom) in SMS-CTR. Δ, trametinib minus DMSO; TPM, Transcripts per Million. **m** Representative ChIP-seq tracks for SNAI2 (blue), MYOD (green), H3K27ac (yellow) (top) and gene expression (RNA-seq) (bottom) in SMS-CTR. Δ, shSNAI2.1 minus shScr; TPM, Transcripts per Million. **n** Western blot of RD, JR1, and SMS-CTR cells treated with 10 nM trametinib compared to vehicle control (DMSO), probed for p-ERK (phosphorylated), ERK, SNAI2, MEF2C, and actin (loading control). Representative blot, n = 3 biologically independent experiments. **o** Schematic model: SNAI2 expression in FN-RMS is regulated by MYOD and SNAI2 binding is able to dampen MYOD binding at myogenic transcription factor genes, which contributes to maintenance of a myogenic differentiation block (left). MEK inhibition with trametinib and/or SNAI2 knockdown in FN-RMS releases SNAI2, allowing MYOD to activate genes important for myogenic differentiation (MYOG, MEFs, TNNTs) thus inducing muscle differentiation (middle). Silencing of myogenic SNAI2 target genes (like MYOG or MEF2A) blocks differentiation downstream of SNAI2. Bubbles depict transcriptional co-activators in inactive (gray) or active (yellow) status.

example, *MYBPH*). Our data can explain why MYOD that is robustly expressed and diagnostic of FN-RMS tumors, can modulate early tumor myoblast-like proliferation but is unable to initiate late myotube-like gene expression that would result in exit from the cell cycle and terminal differentiation when SNAI2 is expressed. However, once *SNAI2* expression is extinguished, MYOD can quickly occupy these enhancer elements and along with *MEF2A/C/D*, *MYOG*, and *CDKN1A* drive robust expression of multiple terminal myogenic structural proteins. Both MEF2 and p21$^{Cip1}$(*CDKN1A*) which unlike MYOD or MYOG are poorly expressed in FN-RMS cells[36,37] when activated together with MYOD and MYOG inhibit tumor growth by driving a terminal differentiation program[38,39]. Our results are consistent with and expand the findings in murine myoblasts using an over-expression analysis of Snai1 binding to demonstrate that during early myogenesis, MYOD is unable to bind muscle differentiation genes with canonical G/C-rich binding sites (E-boxes) that are associated with differentiated genes since they are potentially occupied by Snai1[29]. We previously explored the role of SNAI1 in FN-RMS and while we find overall similar effects on sphere formation, the differentiation effects we observed were not as prominent as SNAI2 knockdown, but more importantly we found that *SNAI1* regulates only *MEF2C*, with relatively no effect on *MEF2A/D*[40]. Conversely, SNAI2 represses MEF2A/C and D expression, suggesting mechanistic differences in the roles of SNAI1 and SNAI2 with respect to FN-RMS differentiation. Thus, our study defines endogenous SNAI2 chromatin binding, currently a challenge in the field, and provides several unique insights on function that are likely to be relevant beyond RMS. Further, since many tissue-specific transcription factors bind E-box elements, and SNAI2 has the ability to bind a wide repertoire of them, albeit with different affinities, understanding the extent to which similar competition with *SNAI2* occurs may potentially determine differentiation state/gene expression in other tissues and or tumors.

MYOD function can also be disrupted by competition for functional E2A co-factor with non-functional E2A splice variants and other expressed factors including TWIST1, TWIST2, and HEY1[9,10,23,41]. However, in the case of TWIST1 and HEY1 this disruption occurs by competition for co-factor E2A and not at the chromatin level[11,22]. Recently, using an inducible over-expression system in a Twist2 positive (Tw2 +) myoblast cell line, Li et al.[41], showed that *Twist2* can activate *Snai2* expression, and can compete with all MYOD binding at E-boxes elements generally inhibiting myogenic gene expression. This report also suggests that Twist2 can influence accessibility to chromatin[41].*TWIST2* that is gained in RMS[41] might be another gene in addition to MYOD that can regulate *SNAI2*. However, while Twist2 appears to generally inhibit myogenesis, our data indicate that SNAI2 downstream of MYOD and RAS/ERK signaling is likely responsible for selective repression of terminal differentiation gene expression through competition with MYOD binding and repression of a *MYOG, MEF2*, and *CDKN1A* differentiation program. Further, the involvement of SNAI2 downstream of RAS/ERK signaling, a driver of FN-RMS, could explain the high dependency of this RMS subtype on SNAI2.

In summary, our study identifies SNAI2 as a core oncogene that blocks terminal differentiation and promotes growth and self-renewal in FN-RMS. SNAI2 regulates the transcriptional output of MYOD and myogenic terminal differentiation regulators *MYOG, MEF2A/C/D*, and *CDKN1A*. Moreover, as SNAI2 depletion mimics blockade of the mutant RAS signal (both at the transcriptional level and phenotypic differentiation level), we propose that SNAI2 connects the genetic and epigenetic drivers of FN-RMS. Finally, our results highlight the importance of SNAIL transcription factors for lineage commitment choices made at the enhancer interface.

## Methods

**Animals**. Mice used for xenograft were CB17 SCID females aged 6–8 weeks. They were maintained at sterile conditions with five mice per cage and fed ad libitum, 12 h light/12 h dark cycle, ambient temperature 18–23 °C with 40–60% humidity. Experimental groups were randomly assigned. Xenograft transplantations were performed in collaboration with Dr. Peter Houghton (Greehey Children's Cancer Research Institute, University of Texas Health Sciences Center, San Antonio, TX, USA) and in collaboration with SAFU Laboratory (Regina Elena National Cancer Institute, Rome, Italy). All mouse experiments were approved by the Institutional Animal Care and Research Advisory Committee of the University of Texas Health Science Center at San Antonio under protocol number 20150015AR and at the Italian Ministry of Health for the Children's Hospital Bambino Gesù/SAFU Institutes under protocol number 514/2015-PR.

**Cell lines**. The human RMS cell lines RD (Female), JR1 (Female), RH-30 (Male), RH-36 (Male), and SMS-CTR (Male)[42] were gifted by Dr. Peter Houghton, GCCRI. RD18 were a gift of Carola Ponzetto, Department of Oncology, University of Turin, Italy[43]. All lines except RD, JR1, RD18, and SMS-CTR were maintained in RPMI supplemented with 10% Fetal bovine serum (VWR) at 37 °C with 5% $CO_2$. RD, JR1, RD18, and SMS-CTR cells were maintained in DMEM supplemented with 10% FBS at 37 °C with 5% $CO_2$. Cell lines were authenticated by genotyping. Differentiation of RMS cells was performed for 3 days in RPMI/DMEM + 2% Horse serum.

**Lentiviral and siRNA knockdown**. Scramble control shRNA and gene-specific shRNAs were delivered on the lentiviral pLKO.1-background vector (Addgene, shSNAI2.1 #10904, shSNAI2.2 #10905) and packaged using transfected (FuGENE6, Promega) 293T cells. RMS cells were infected with viral particles for 24 h at 37 ˚C with 4 μg/mL of polybrene (EMD Millipore). Gene-specific or control siRNAs were delivered using Lipofectamine transfection reagent (Life Technologies) in flat, clear-bottom 6-well plates. Cell growth and viability was assessed using Incucyte imaging platform. Three independent experiments were carried out.

**Gene expression**. Total RNA from cells was isolated using Qiagen RNA easy kit and cDNA synthesis was performed using High Capacity cDNA Synthesis Kit (ThermoFisher #4368814), quantitative real time PCR was performed on ABI QuantStudio6 Real-Time PCR system. List of PCR primers used in this study is listed in Supplementary Table 2. Fold change gene expression was calculated by $^{\Delta\Delta}$ct method. Significance was calculated by Student's two-tailed *t*-test. Triplicate assays were carried out in three independent experiments.

**Western blotting**. Total cell lysate was obtained by lysing in 1x RIPA buffer (Millipore). Membranes were developed using ECL reagent (Immobilon, Millipore). Post primary antibody blotting, membranes were striped, rinsed, and re-probed with the respective internal control antibodies and imaged. Antibodies used in the study is listed in Supplementary Table 2.

**Immunofluorescence Staining**. Cells were fixed after 3 days in differentiation medium in 4% paraformaldehyde (PFA)/PBS, permeabilized in 0.5% Triton X-100/PBS, and incubated with rabbit α-MEF2C (CST, 5030) and α-myosin heavy chain (DSHB, MF20) in 2% goat serum/PBS. Secondary antibody detection used Alexa-488 goat α-mouse and Alexa-594 goat α-rabbit (Invitrogen). Cells were counter-stained with DAPI and imaged using the Olympus microscope FV3000 with Olympus FV315S-SW image acquisition software. Images were processed in ImageJ and Adobe Photoshop.

**Colony formation assay**. After 72 h of selection a total of $10 \times 10^2$ shScr and shSNAI2 cells were seeded in 6-well plates with 2 mL of DMEM (10% FBS). Medium was refreshed every 2 days, and after 14 days, cells were fixed and stained with Diff-Quik® (Medion Diagnostic AG460.053) as per manufacturer's instruction. Colonies containing >50 cells were counted. Triplicate assays were carried out in three independent experiments.

**Soft agar colony formation assay**. After 72 h of selection a total of $10 \times 10^3$ shScr and shSNAI2 cells were suspended in DMEM (10% FBS) containing 0.35% agar (NuSieve-GTG-Agarose). Cells were seeded on a layer of 0.7% agar in DMEM (10% FBS) in 6-well plates. Media was refreshed every 2 days. On week 4, colonies were counted by microscopic inspection. Colony numbers were normalized by dividing the number of colonies by the number of total units (colonies + single cells). Triplicate assays were carried out in three independent experiments.

**Sphere formation assay**. Serially diluted RD, JR1, or SMS-CTR cells were resuspended in 100 μL Neuro Basal Medium (NBM)[44]. The diluted cells were gently added to 1 mL of pre-warmed NBM per well of a 24-well plate (Corning Cat#3603). The cells were incubated for at least 10–12 days with 100 μL of fresh media added every 2 days. The spheres were counted manually or using the Celigo

Imaging Cytometer automated cell counting apparatus (Nexcelom Bioscience LLC, Lawrence, MA, USA).

**Clinical specimens.** Primary samples were obtained from the Pathology Unit of Bambino Gesù Children's Hospital (Rome, Italy) and Department of Pathology, University of Washington (Seattle, WA, USA) and the approval of the study was obtained from the ethics committees at the two Research Centers. Written informed consent was obtained from all patients.

**Immunohistochemistry.** Immunohistochemistry was performed on 2 μm-thick sections obtained from formalin-fixed tissue embedded in paraffin. After dewaxing and rehydrating, heat-induced epitope retrieval was performed by boiling the slides with EDTA (pH 9) (Dako, Glostrup, Denmark). Endogenous peroxidase was blocked with 3% hydrogen peroxide followed by incubation with mouse-to-mouse blocking reagent to inhibit endogenous mouse immunoglobulin and then another blocking with blocked BSA 5%. Sections were incubated overnight at +4 °C with mouse monoclonal α-MF20 antibody (dilution 1:50; DSHB) or rabbit monoclonal α-SNAI2 (dilution 1:100). Detection of the primary antibody was performed by using the appropriate secondary biotinylated antibody (Dako, Carpinteria, USA) and the peroxidase DAB kit (Dako, Carpinteria, USA) with or without counterstaining with Gill's hematoxylin (Dako, Carpinteria, USA). Negative controls were stained in parallel with either isotype non-specific IgG or only the primary antibody. The light microscopy imaging was performed on a Nikon E600 light microscope equipped with NIS Elements BR software, using 20x objective.

**RNA-seq.** RNA was extracted using the RNeasy mini kit (Qiagen). Poly-A selected RNA libraries were prepared and sequenced on Illumina HiSeq2000. QC was performed using FastQC version 0.11.2 and Picard's version 1.127 RNASeqMetrics function with the default parameters. PCR duplicates were marked using Picard's MarkDuplicates function. RNA-seq reads were aligned to the UCSC hg19 reference genome using TopHat version 2.0.13. Significance was defined as having FDR $q = $ <0.01 and FWER $p ≤ 0.05$. Gene set enrichment analysis (http://www.broadinstitute.org/gsea/index.jsp) was performed using default parameter settings.

**ChIP-seq.** Formaldehyde-fixed (1%, 12 min) cells (SMS-CTR, RD, and JR1), were sheared to achieve chromatin fragmented to a range of 200–700 bp using an Active Motif EpiShearSonicator. Chromatin samples were immunoprecipitated overnight at 4 °C with antibodies targeting SNAI2 (CST, Catalogue # 9585), MYOD (sc, Catalogue #760), and H3K27ac (Active Motif, cat. #39133). DNA purifications were performed with the ChIP-IT High Sensitivity kit (Active Motif). We employed ChIP-seq spike in using Drosophila chromatin (Active Motif, Catalogue #53083) and an antibody against Drosophila-specific histone variant H2Av (Active Motif, Catalogue #61686).

ChIP-seq libraries were prepared using Illumina TruSeqChIP Library Prep Kit (Illumina). Libraries were multiplexed and sequenced using the NextSeq500 (Illumina). Reads were mapped to reference genome (version hg19) using BWA[45]. High-confidence ChIP-seq peaks were called by MACS2.1[46]. Peaks from ChIP-seq of MYOD and H3K27ac were selected at a stringent $p ≤ 0.0000000001$, while a less restrictive $p ≤ 0.000001$ was applied to SNAI2. Gene ontology was performed using GREAT, using hg19 and the whole genome as the background. Chromatin states were characterized using ChromHMM (http://compbio.mit.edu/ChromHMM/). Enhancers were identified using the ROSE2 (https://github.com/linlabbcm/rose2) software. Differential peak calling was performed using BEDTools v2.25.0 in multicov mode to measure read counts, which were normalized per million mapped reads, and visualized in R package ggplot2 or NGS plot. Enhancer regions were linked to their nearest gene, irrespective of strand specificity and gene direction, within topologically associated domain (TAD) boundaries using EDEN. Enrichment of known and de novo motifs were found using HOMER. Enrichment peaks were visualized with epigenome browser[47] and IGV[48]. Raw sequencing data and processed files are available (GEO GSE137168).

**HiChIP.** SMS-CTR cells were fixed with DSG for 10 min at room temperature (23 °C), then 1% formaldehyde for 12 min at room temperature. Cells (~8 million) were then lysed gently to release nuclei, permeabilized in 0.5% SDS for 10 min at 62 °C, quenched with 10% Triton X-100, and digested with DpnII (400 U, overnight at 37 °C) which was then heat inactivated (20 min, 62 °C). Biotin incorporation was done with biotin-14-dATP (Thermo, Cat# 19524-016) and DNA Polymerase I, Large (Klenow) Fragment (NEB, Cat# M0210) for 1 h at 37 °C. Then we performed in situ ligation with T4 DNA ligase (2 h room temperature and 16 °C overnight). Nuclei were pelleted and sonicated (28 cycles with shearing 'on' time with 30 s 'on' 30 s 'off', using the Active Motif Epi-shear probe sonicator, 30% power). Lysates were immunoprecipitated with anti-H3K27ac (Active Motif, Cat# 39133), overnight incubation (rotating at 4 °C), then bound to Dynabeads Protein A (Thermo, Cat# 10002D) and incubated (2 h at 4 °C). Dynabeads were then held magnetically, washed, eluted, treated with Proteinase K (30 min at 55 °C); crosslinks were reversed by heating to 67 °C for 2 h. DNA was purified using ChIP DNA Clean & Concentrator kit (Zymo Cat# D5205). Biotin capture and washing was done with Dynabeads M-280 Streptavidin (Thermo, Cat# 11205D), followed by end repair, A-tailing, adapter ligation, and library amplification all on-bead as

previously reported[49]. Libraries were paired-end sequenced to a shallow depth of 80–120 million reads. Two independent biological replicates for SMS-CTR were combined informatically to get contact maps with 30 million valid, long-range *cis* contacts. Analysis was performed using HiC-Pro[50] and visualized in Juicebox[51].

**dCas9-KRAB of SNAI2 regulatory elements.** Guide RNA sequences were designed using the BROAD Institute GPP sgRNA design tool (https://portals.broadinstitute.org/gpp/public/) and cloned into the BsmBI restriction sites of pLV hU6-sgRNA hUbC-dCas9-KRAB-T2a-Puro lentiviral plasmid (Addgene #71236). gRNA target sequences are listed in Supplementary Table 2. Lentiviral supernatants were generated using Lenti-X 293 T (Takara Bio) transfected with psPAX2-Gag/Pol (Addgene #12260) and pMD2.G-VSVG (Addgene #12259) using Viafect Transfection Reagent (Promega). SMS-CTR cells were transduced in triplicate with lentiviral supernatant supplemented with polybrene (8 μg/mL final). At 24-h and 48-h post-transduction, transduced SMS-CTR cells were selected with 1 μg/mL puromycin. RNA was isolated at 72-h post-transduction using the RNeasy Plus Mini Kit (Qiagen Cat. #74134). cDNA was prepared from 500 ng of total RNA using the SuperScript First-Strand cDNA Synthesis Kit (Invitrogen Cat. #11904-018). Real-time PCR was performed using FastStart Universal SYBR Green Reagent (Sigma Cat. #4913850001) using a ViiA 7 Real-Time PCR System (ThermoFisher). Primer sequences for detection of SNAI2 and GAPDH are reported in Supplementary Table 2. SNAI2 expression was calculated using the $2^{-ΔΔCT}$ method and statistical analysis between groups was performed using *t*-test with Welch's correction.

**Re-ChIP.** SMS-CTR cells were fixed and sonicated as with ChIP-seq experiments. The first ChIP was performed with MYOD antibody (sc, Catalogue #760) and incubated with sheared chromatin, protease inhibitor cocktail, and Protein G magnetic beads (Active Motif Re-ChIP-IT kit) at 4 °C overnight. Magnetic beads were washed, and chromatin was eluted and desalted according to manufacturer's instructions. LSV Protein Magnetic beads were used for the second ChIP reaction, employing SNAI2 antibody (CST, Catalogue #9585). In the same way, the reverse sequence was also performed (first ChIP SNAI2, then MYOD). DNA eluted was purified, and qRT-PCR was performed on ABI QuantStudio6 Real-Time PCR system. List of PCR primers used in this study is listed in Supplementary Table 2. Enrichment of DNA was quantified by the standard curve method.

**Quantification and statistical analysis.** The significance of results was assessed by applicable statistical tests for each experiment described. Except where stated all experiments were performed at least in three biological replicates. Student's two-tailed *t*-test was used for testing significance of qRT-PCR, sphere assays, colony assays, soft agar assays, and immunohistochemistry staining values and exact *p* values were reported on the related figures or in the text. Murine experimental data was tested using 2-way ANOVA or two-tailed *t*-test, a minimum of three mice was used per condition. Representative imaging data from three biological replicates was used to quantify immunofluorescence staining data and significance was assessed by Student's two-tailed *t*-test and exact *p* values were reported on the related figures or in the text. Statistical analysis of NGS data was performed as per the analysis program default parameters or else as described in the NGS section. Software used for statistical tests include R, Graphpad Prism, and Microsoft Excel.

**Reporting summary.** Further information on research design is available in the Nature Research Reporting Summary linked to this article.

## Data availability

The datasets generated during this study (RNA-seq, ChIP-seq, and Hi-seq) are available at GEO (Gene Expression Omnibus) with the following accession number (GSE137168). Previously published GEO datasets used in the study were: ChIP-seq in human myoblasts and myotubes (GSE29611, GSE50413), ChIP-seq in RMS and ChIP-seq and RNA-seq in Fibroblasts and Fibroblasts + *PAX3-FOXO1* (GSE83728), RNA-seq in Fibroblasts and Fibroblasts + *MYOD1* (GSE93263), Hi-C in IMR90 (GSE63525), ChIP-seq and RNA-seq in SMS-CTR ± Trametinib (GSE85171). The remaining data are available within the article, supplementary Information or available from the authors upon request.

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

## Acknowledgements

The Myogenin (Wright WE) and MHC (Fishman DA) antibodies were obtained from the Developmental Studies Hybridoma Bank, developed under the auspices of the NICHD and maintained by The University of Iowa, Department of Biology, Iowa City, IA 52242, USA. Grant Support: This project has been funded with federal funds from NIH grants MI and PH (R00CA175184, NCI P01 CA165995), CPRIT grant (RR160062) to MI, Associazione Italiana Ricerca sul Cancro (AIRC 15312) and Italian Ministry of Health (PE-2013-02355271) to RR, AIRC 5xmille (9962) to FL. S. Pomella is a recipient of a Fondazione Veronesi fellowship. MI is a recipient of the Max and Minnie Tomerlin Volker fund young investigator award. Kunal Baxi is a T32/TL1 fellow (CA148724) and (TL1TR002647).

## Author contributions

S.P., P.S., B.E.G. performed and/or interpreted or supervised aspects of the different experiments and wrote the manuscript. B.E.G. designed and implemented scripts, pipelines and analysis tools for RNA-seq and ChIP-seq. H.-C.C. built Hi C pipeline and analyzed HiChIP data. P.S., E.C. initiated the experimental study. L.W., N.H., K.B., M.C. helped with shRNA and siRNAs silencing experiments. D.M. designed guide RNAs, built plasmids, and performed dCas9-KRAB enhancer disruption experiments. E.C., C.D.S., and R.D.V. helped with immunohistochemistry and pathological analyses. L.W., P.H., B.A., and I.C. helped with animal experiments. Y.S. performed library preparation for ChIP-seq experiments. M.E.Y. assisted with data interpretation. B.Z.S. supervised HiChIP experiments. Y.C. performed QC and initial analysis of RNA-seq data. P.H., F.L. discussed the clinical importance of the results and provided funding. M.I., R.R., and J.K. in coordination, provided overall study direction, funding, supervision, and revised the manuscript. All authors critically reviewed the report and approved the final version.

## Competing interests

The authors declare no competing interests.
