## [Peer review file · Nature Communications]

Reviewers' comments:

Reviewer #1 (Remarks to the Author):

Herein, Pomella and colleagues describe how SNAI2 may preferentially function in fusion-negative rhabdomyosarcoma (FN-RMS) as an oncogene through a multi-faceted mechanism. RMS represents a classic embryonal tumor of childhood presents as an arrested state of skeletal muscle development. These tumors express the fate determining transcription factor MYOD1 and differentiating transcription factor MYOG but fail to complete cellular fusion and differentiation in to mature skeletal muscle. The keys determinants that block the activity and normal differentiation program downstream of MYOD1 remains unknown. In this manuscript Pomella et al. aim to elucidate the role of SNAI2 in this process. The authors describe SNAI2's role in promoting growth and blocking myogenic differentiation, a histological hallmark of FN-RMS. These phenotypes can be attributed to SNAI2 outcompeting MYOD1 off of myogenic enhancer elements associated with terminal myogenic differentiation. When SNAI2 was removed from their model system, the competition with MYOD1 was lost and myogenic differentiation was restored. These results are intriguing and can partially explain the paradoxical high MYOD1 expression/low myogenic differentiation observations seen in FN-RMS. This has been an open question in the field and this manuscript does provide strong data to reconcile this paradox. The manuscript was very well-written, concise and beautifully illustrated. The bioinformatics analyses are a strength of the paper. However, there are a few major issues that must be addressed in order to warrant publication in Nature Communications, mainly in regard to their cell line usage and in vivo studies as detailed below:

Major Concerns:

- 1) Throughout the manuscript, there is inconsistent use of cell lines. For example, SMS-CTR used in Figure 1H, RD throughout Figs. 2-3, SMS-CTR in Figure 4B and C, etc. Sometimes the other cell line was used in the Supplement, sometimes not, and if there, the results are not as striking. Even if unintentional, it does give pause to the reader that the data is cherry-picked. Consistent cell line usage throughout the manuscript is needed. For example, in Figure 3, use SMS-CTR xenografts as most of the mechanistic enhancer work is done with the SMS-CTR cells. Or use RD cells to recapitulate Figure 2F where JR1 cells were used alone.
- 2) Dovetailing with comment 1, it appears that the JR1 cell line is a subclone of RD cells (Zheng...Skapek, MCR 2019). Xenograft experiments would need to be completed with a different cell line, such as SMS-CTR, to confirm the result. Either that or show via STR profiling that the lines are distinct. The SMS-CTR xenografts should probably be done either way as the xenograft experiment results seem to have marginal differences (without vincristine). Although the data are statistically significant, 3 replicates for the test xenografts (Figure 3G-I) seems under-powered. The authors should perform a power calculation to determine the number of animals used was adequate for the phenotypic differences observed.
- 3) In general, the detail in the methods is scant about number of replicates per experiment and number of times experiments were repeated. Please provide that information. For example, in Figure 1C, no detail was given about number of human RMS samples in the cohort, where they were obtained, or how many were positive for SNAI2. A negative control slide would be pertinent to show that the staining depicted is not just off-target IHC staining.
- 4) In Figure 1I, the dependency of SNAI2 in FN-RMS as reported in the manuscript from Project

Achilles data was not apparent when re-evaluated at review with the newest build of the DepMap (19Q4). In fact, JR1 has a positive CERES dependency score indicating no dependency and SMS-CTR has a score greater than -0.5 indicating no significance. Please specify which build of the DepMap was used to generate this Figure as the reviewer is concerned that this may not be correct. That said, the lack of correlation with the DepMap does not discount the biological phenotypes reported in this paper.

5) In Figure 2G-J, the sphere forming capacity reduction is striking. Are there any changes in expression of stemness-related genes? One could suggest that this is merely a byproduct of decreased proliferation.

Minor Comments

- 1) In the abstract, not all FN-RMS tumors have RAS pathway activating mutations. Please revise wording to include this fact.
- 2) Alveolar and embryonal RMS are not always FP- or FN-RMS, respectively. Rephrase to address this in the first paragraph of the introduction.
- 3) It is unclear what the author's mean when they call PAX7 a precursor marker (a few times throughout the manuscript). Do they mean satellite cell? Clarification on this is needed.
- 4) Unsure if I see the data for SNAI2 inhibiting the expression of MEF2C or CDKN1A as mentioned in the last sentence of the next to last Results section.

Reviewer #2 (Remarks to the Author):

In this manuscript, the authors found that SNAI2 blocks terminal differentiation and promotes growth and self-renewal in FN-RMS. Mechanistically, SNAI2 regulates the transcriptional output of MYOD and myogenic terminal differentiation regulators MYOG, MEF2A/C/D and CDKN1A. Although some of these findings are interesting, the major conclusion is premature. Several major concerns are indicated below:

1. The authors claimed that MYOD1 positively regulate SNAI2 transcription through binding to SNAI2 super enhancers in FN-RMS cells, but the inhibition of MYOD1 does not repress SNAI2 transcription remarkably (Fig 1H), especially in RD cell (Fig S1B). While ectopic expression of MYOD1 dramatically up-regulated MYOG expression (Fig S1C), its effect on SNAI2 is minor. It is also noticeable that though MYOD1 level is much higher in 48h-siScr group, SNAI2 level is somehow slightly lower than 24h-siScr group in Fig S1B. These results raise the question that if MYOD1 is the major regulator of SNAI2 expression. In addition, though HiC and H3K27ac ChIPseq results indicate highlighted regions in Fig 1E might be super enhancers, more evidence is needed to prove that MYOD1 regulate SNAI2 through binding to these super enhancers.

2. Does MYOD1 also regulate SNAI2 in FP-RMS in the same manner? Will suppression of SNAI2 activate myogenic differentiation in FP-RMS? What might be the cause that SNAI2 is only dependent in FN-RMS but not in FP-RMS (Fig 1I)?

3. In order to prove the co-occupancy of SNAI2 and MYOD1, ChIP-re-ChIP assay is required to strengthen this conclusion.

4. In Fig 5F, more evidence is required to prove the highlighted regions are bona fide enhancers of these myogenic genes.

5. The mechanism of how SNAI2 co-binding to the MYOD1 sites suppresses myogenic gene transcription has not clearly defined. The regulatory regions of the SNAI2 repressed myogenic genes still show strong H3K27ac signal (Fig 4A, 4D and 5F); what is the chromatin accessibility of these regions?

Minor concerns:

1. In Fig 3P, the error bar for the second shScr group is missing.
2. In Fig S3S, the numbers of mice in each group vary greatly, what is the rationale for that design?
3. The colors are too similar to be distinguished among groups in Fig 4D and 5D.

Reviewer #3 (Remarks to the Author):

The paper by Pomella et al identifies the transcription factor SNAI2 as oncogene in fusion-negative rhabdomyosarcoma. It is convincingly shown that reduction of SNAI2 levels induces differentiation both in vitro as well as in vivo by a variety of cell biological assays in three different cell lines. Mechanistically, the authors identify enhancers bound by SNAI2 by ChIPseq experiments acting in competition to MYOD binding at E-boxes of differentiation genes. In addition, SNAI2 suppresses the myogenic program via down-regulation of MYOG, MEF2A and CDKN1A. The findings are novel and advance our understanding of the differentiation block found in rhabdomyosarcoma despite expression of myogenic master regulators.

Minor points:

- The abstract lacks a concluding sentence
- Figure 1B: SNAI2 expression in Rh18 cells is very low. Does this also result in a differentiation block?
- Supplemental Figure 1C lacks statistics
- Supplemental Figure 2H, what are RD18 cells? They are not mentioned in Materials.
- Figure 4, S4 and following: In Figure S4, the authors demonstrate that only 146 SNAI2 binding peaks are common to all three cell lines analyzed. It is unclear whether go analysis in 4B refers to these overlapping sites/genes or not. If not, a GO analysis specifically for these overlaps should be shown. Further, Figure 4 talks about 664 sites overlapping between SNAI2 and MYOD in all 3 cell lines (legend). These numbers are not consistent.
- Figure 5C: Differences in MYOD ChIPseq at MYOD sites appear very minor, Is this really significant?

- Figure 6A: The scheme should not have an activating arrow from "SNAI2" to "myogenic TF activated", as is rather blocks differentiation.

Major Concerns:

1) *Throughout the manuscript, there is inconsistent use of cell lines. For example, SMS-CTR used in Figure 1H, RD throughout Figs. 2-3, SMS-CTR in Figure 4B and C, etc. Sometimes the other cell line was used in the Supplement, sometimes not, and if there, the results are not as striking. Even if unintentional, it does give pause to the reader that the data is cherry-picked. Consistent cell line usage throughout the manuscript is needed. For example, in Figure 3, use SMS-CTR xenografts as most of the mechanistic enhancer work is done with the SMS-CTR cells. Or use RD cells to recapitulate Figure 2F where JR1 cells were used alone*

We thank the reviewer for the feedback and agree that there should be consistent cell line usage. We have revised the manuscript and now show robustness of our findings in multiple RD, SMS-CTR and JR1 cell lines, and experimental analyses. We have also reorganized Figure panels to include all data from a single cell line in a given figure with supplemental figures showing supporting complementary data from other cell lines and *in vivo* experiments to show robustness of our findings.

For example,

- We have reorganized Figure 2 and 3: all panels now show only RD experiments with new experiments performed in panel 2f with RD cells.
- Supplemental Figure S2-S5 show reproducibility of the data in the other cell lines: JR1, SMS-CTR and RD18 cells. They contain new experiments in SMS-CTR cells (Figure S2 i-p, Figure S3 g-i and Figure S3 l, Figure S5 a-m). Experiments in FN-RMS SMS-CTR cells include *in vitro* experiments silencing SNAI2 and assessing proliferation, soft agar, colony formation, sphere forming abilities, effect on differentiation by analyzing expression of myogenic genes (qRT-PCR) and proteins (western blot and immunofluorescence staining). We also performed *in vivo* xenograft experiments on SMS-CTR cells with two different SNAI2 shRNAs (Figure S5 f-m).
- Figure 6n, we have added a new panel showing that MEK inhibition with Trametinib in SMS-CTR cells results in loss of phosphoERK, SNAI2 knockdown, and induction of MEF2C expression.

2) Dovetailing with comment 1, it appears that the JR1 cell line is a subclone of RD cells (Zheng...Skapek, MCR 2019).

In the publication by Zheng et al. (2019) RD-like cells are mentioned and not the JR1 cell line, which is cytogenetically and molecularly different from the RD cell line (Sokolowski et al. *Oncogene*. 2014;33(15):1877-1889. doi:10.1038/onc.2013.129; Hinson et al. *Front Oncol*. 2013;3:183. doi:10.3389/fonc.2013.00183). In line with this, STR analysis performed by Dr. Peter Houghton's laboratory indicates that RD and JR1 are not identical containing distinct differences (personal communication with Dr. Peter Houghton).

3) Xenograft experiments would need to be completed with a different cell line, such as SMS-CTR, to confirm the result. Either that or show via STR profiling that the lines are distinct. The SMS-CTR xenografts should probably be done either way as the xenograft experiment results seem to have marginal differences (without vincristine). Although the data are statistically significant, 3 replicates for the test xenografts (Figure 3G-I) seems under-powered. The authors should perform a power calculation to determine the number of animals used was adequate for the phenotypic differences observed.

As requested, we have performed *in vivo* xenografts experiments on SMS-CTR cells with two different SNAI2 shRNAs using 7 mice per treatment group and show that indeed SNAI2 is required for growth *in vivo* (Figure S5 f-m).

In our initial submission we injected RD cells in 6 shScr mice on the left leg and 3 of them with either shSNAI2-1 or shSNAI2-2 in the right leg to reduce the number of mice having, at the same time, sufficient power calculation (6 control samples and 6 SNAI2-depleted samples). However, we do agree with the reviewer that for each shRNA, 3 shSNAI2 xenografted mice used though significant is underpowered; hence we provide new *in vivo* experiments in SMS-CTR cells.

3) In general, the detail in the methods is scant about number of replicates per experiment and number of times experiments were repeated. Please provide that information.

We thank the reviewer for pointing out this oversight on our part. We have updated the methods section to clearly indicate biological replicates and additional experimental details. In general, all *in vitro* cell culture based experiments were repeated in triplicate or more repeats. Additionally, the experimental conclusions obtained from analyses use two or more well established RMS cell lines. We have added new experiments in SMS-CTR both *in vitro* and *in vivo*. Together our revised data greatly strengthen our conclusions.

For example, in Figure 1C, no detail was given about number of human RMS samples in the cohort, where they were obtained, or how many were positive for SNAI2. A negative control slide would be pertinent to show that the staining depicted is not just off-target IHC staining.

The main text has been revised to include the relevant information: 19 FN-RMS (14 were positive and 5 were negative for SNAI2 staining) and 4 FP-RMS (3 positive and 1 negative for SNAI2) were assessed by IHC. The text, methods and main and supplemental figures have been revised to report negative controls. Furthermore, as requested, we provide muscle tissue staining for SNAI2 as a negative tissue control. In addition, RMS tissues stained using either primary isotype antibody or only the secondary antibody were also used as controls for the staining (Figures 1c and S1 b). Dr. Eleanor Chen (co-author), a soft tissue pathologist with expertise in Rhabdomyosarcoma assisted us with the histological analyses.

4) In Figure 1I, the dependency of SNAI2 in FN-RMS as reported in the manuscript from Project Achilles data was not apparent when re-evaluated at review with the newest build of the DepMap (19Q4). In fact, JR1 has a positive CERES dependency score indicating no dependency and SMS-CTR has a score greater than -0.5 indicating no significance. Please specify which build of the DepMap was used to generate this Figure as the reviewer is concerned that this may not be correct. That said, the lack of correlation with the DepMap does not discount the biological phenotypes reported in this paper.

We have revised and updated Figure 1i, and moved this data to Figure S1 g using newly released 20Q2 data from DepMap, also showing additional details of the RMS cell lines analyzed in DepMap in Figure S1 h. Consistent with our earlier figure the DepMap data show that FN-RMS cell lines are more sensitive to SNAI2 perturbations compared to FP-RMS cell lines. We would like to point out that the JR cell line reported in the DepMap is not the same as FN-RMS JR1 but rather is a FP-RMS (Sokolowski et al. *Oncogene*. 2014;33(15):1877-1889. doi:10.1038/onc.2013.129; Hinson et al. *Front Oncol*. 2013;3:183. doi:10.3389/fonc.2013.00183).

5) In Figure 2G-J, the sphere forming capacity reduction is striking. Are there any changes in expression of stemness-related genes? One could suggest that this is merely a byproduct of decreased proliferation.

As suggested, we performed qRT-PCR as well as RNA-seq comparing expression in spheres from shScr and shSNAI2 RD cells and have observed a loss of expression in stemness markers *NANOG*, *OCT4*, *SOX2* and *VANGL2* in SNAI2 knock-down spheres. These results are shown in Ancillary Figure 1A-B for the Reviewer. We have not included this in the manuscript as it detracts from the main thrust of the manuscript.

Minor Comments

1) *In the abstract, not all FN-RMS tumors have RAS pathway activating mutations. Please revise wording to include this fact.*

We have incorporated suggested change.

2) *Alveolar and embryonal RMS are not always FP- or FN-RMS, respectively. Reword to address this in the first paragraph of the introduction.*

We have reworded our text to clarify and incorporated suggested change.

3) *It is unclear what the author's mean when they call PAX7 a precursor marker (a few times throughout the manuscript). Do they mean satellite cell? Clarification on this is needed.*

We have reworded our text to say "PAX7, a marker of undifferentiated satellite muscle stem cells"

4) *Unsure if I see the data for SNAI2 inhibiting the expression of MEF2C or CDKN1A as mentioned in the last sentence of the next to last Results section.*

Our data show that *MEF2C* and *CDKN1A* are likely indirect *SNAI2* targets. This is supported by the following evidence

- 1) The repressive effects of SNAI2 on MEF2C are reported in Figure 2b-e (for RD cells) and Figure S2 b-e (for JR1) by immunofluorescence and qRT-PCR. New data have been added for SMS-CTR in Figure S2 j-m.
- 2) The repressive effects of SNAI2 on CDKN1A/ p21^{Cip1} data in RD are shown in Figure 2e and 2f, and in RD18 and JR1 cells in Figure S3 a, d, j, k. New data have been added for SMS-CTR in Figure S3 g, l.
- 3) Finally, we provide additional ChIPseq data for SNAI2, MYOD and H3K27ac binding at the genomic loci of *MEF2C* and *CDKN1A* showing that while MYOD and H3K27ac binding increases at the loci, yet SNAI2 is not bound in these genomic regions (Ancillary Figure 2 for the Reviewer) indicating that SNAI2 indirectly regulates *MEF2C*, and *CDKN1A* expression.

Reviewer #2 (Remarks to the Author):

In this manuscript, the authors found that SNAI2 blocks terminal differentiation and promotes growth and self-renewal in FN-RMS. Mechanistically, SNAI2 regulates the transcriptional output of MYOD and myogenic terminal differentiation regulators MYOG, MEF2A/C/D and CDKN1A. Although some of these findings are interesting, the major conclusion is premature. Several major concerns are indicated below:

1a) *The authors claimed that MYOD1 positively regulate SNAI2 transcription through binding to SNAI2 super enhancers in FN-RMS cells, but the inhibition of MYOD1 does not repress SNAI2 transcription remarkably (Fig 1H), especially in RD cell (Fig S1B).*

1b) *While ectopic expression of MYOD1 dramatically up-regulated MYOG expression (Fig S1C), its effect on SNAI2 is minor. It is also noticeable that though MYOD1 level is much higher in 48h-siScr group, SNAI2 level*

is somehow slightly lower than 24h-siScr group in Fig S1B. These results raise the question that if MYOD1 is the major regulator of SNAI2 expression.

We would like to thank the Reviewer for his questions about the data in Figure 1h, Figure S1 b and S1 c pertaining to the question that *MYOD1* is a major regulator of *SNAI2* expression in FN-RMS. We went back and reexamined our primary data and performed additional experiments to address the questions raised.

1 a and b) First, we went back to our blots and qRT-PCR raw data and realized that the data at 48h post-siRNA transfection were not correctly normalized. Thus, we replaced the qRT-PCR graphs normalizing *MYOD1* and *SNAI2* expression levels on the respective control at the right time point (24 and 48h) (Figure 1h and Figure S1 d). This led to a clearer difference between sh*SNAI2* and shScr samples. Additionally, we show the three biological replicates (N=3 films) that demonstrate that *SNAI2* protein levels are reduced after siRNA knockdown of *MYOD* in RD cells in Ancillary Figure 3A for the Reviewer.

1 a and b) In a second set of experiments, we knocked out *MYOD1* by a CRISPR approach to show that *SNAI2* at both mRNA and protein level was reduced following *MYOD1* deletion (Ancillary Figure 3B for the Reviewer).

1c) In addition, though HiC and H3K27ac ChIPseq results indicate highlighted regions in Fig 1E might be super enhancers, more evidence is needed to prove that MYOD1 regulates SNAI2 through binding to these super enhancers.

1c) We thank the reviewer for this comment. We therefore performed a third set of experiments to test if the *MYOD* bound super enhancers in *SNAI2* are able to modulate *SNAI2* expression. We therefore performed HiChIP and sequencing for H3K27ac which marks super enhancers we show in SMS-CTR cells the *MYOD* bound super enhancers are within the same TAD domain as in IMR-90 cells, see Figures 1d, g.

Second, using a CRISPR/Cas9 defective vector fused to a KRAB transcriptional repressor domain and selecting for Guide RNAs that recognize the *MYOD* binding motifs in enhancers E1-E5 in *SNAI2*, we show that binding of the repressor at the proximal enhancers E1 and E2, significantly repress *SNAI2* expression while enhancer E3 has no effect on *SNAI2* expression (Figure 1i).

2. Does MYOD1 also regulate SNAI2 in FP-RMS in the same manner? Will suppression of SNAI2 activate myogenic differentiation in FP-RMS? What might be the cause that SNAI2 is only dependent in FN-RMS but not in FP-RMS (Fig 1I)?

We have not extensively explored effects of *SNAI2* on differentiation of FP-RMS cells. One of the challenges in the field is that in FP-RMS a clear differentiation hierarchy has not been established. Nevertheless, we have assessed the question raised. We demonstrate that knocking down *MYOD* using siRNA reduces *SNAI2* levels at protein and mRNA level in two of three FP-RMS cell lines (SCMC and RH30) but in RH4 cells a slight reduction is evident 48h after siRNA transfection only at the protein level (Ancillary Figure 3C-E for the Reviewer).

In another set of experiments on RH30 and RH41 FP-RMS cells with stable knockdown of *SNAI2* using two shRNAs (Ancillary Figure 4A-H for the Reviewer), we show that when the cells are grown in differentiation medium (see Methods section) and assessed for differentiation by immunofluorescence staining for MEF2C and myosin MyHC, RH30 cells show a very slight effect on differentiation only with one shRNA, while RH41 cells have a modest increase in the differentiation markers. Altogether, these results suggest that *SNAI2* depletion can induce a differentiated phenotype in a FP-RMS cell line-dependent manner.

We also assessed the differentiation in RH18, an additional FN-RMS cell line showing a significant and robust increase in myogenic differentiation similar to all other FN-RMS cell lines tested (Ancillary Figure 4I-L for the Reviewer). Moreover, as reported in response to point 1) of Reviewer #1, SMS-CTR FN-RMS cells have been also been tested similarly in new experiments and show robust myogenic differentiation (Figure S2 j-m, Figure S3 g-i and l).

Together, our data show that in a subset of FP-RMS cells, MYOD can maintain *SNAI2* expression and *SNAI2* suppression may lead to modest differentiation, at least compared to FN-RMS. Considering that FP-RMS and FN-RMS are two RMS subtypes with distinct molecular landscapes, these new data could be deeper explored in future studies.

3. In order to prove the co-occupancy of SNAI2 and MYOD1, ChIP-re-ChIP assay is required to strengthen this conclusion.

We agree with the Reviewer and performed ChIP-re-ChIP assay on SMS-CTR cells and demonstrate that *SNAI2* and MYOD co-occupy at the *MYOG* SEs. We have added this supporting experiments to the main Figure 4e.

4. In Fig 5F, more evidence is required to prove the highlighted regions are bona fide enhancers of these myogenic genes.

We agree with the reviewer and have generated Hi-ChIP data for H3K27ac in SMS-CTR cells and show that the peaks we designate as enhancers in *MYOG* and *MEF2A* are within the defined TAD regulatory domains, thus supporting our assertion these are regulator super enhancer elements. We present these new experimental data in the revised Figure 5f.

5. The mechanism of how SNAI2 co-binding to the MYOD1 sites suppresses myogenic gene transcription has not clearly defined. The regulatory regions of the SNAI2 repressed myogenic genes still show strong H3K27ac signal (Fig 4A, 4D and 5F); what is the chromatin accessibility of these regions?

We agree with the Reviewer's comment that *SNAI2* does not initiate H3K27ac at enhancers but rather *SNAI2* binding and competition with MYOD at enhancers results in a net inhibition of transcription. Thus, in a *SNAI2* knockdown condition, when only MYOD has access to the gene regulatory regions, MYOD and H3K27ac binding increase and accessibility to chromatin could increase too. In support of this we find that 72 hours of treatment with the MEK inhibitor Trametinib, which induces strong myogenic differentiation in FN-RMS (Ref. 21), in SMS-CTR cells results in the loss of *SNAI2* binding, an increase in MYOD occupancy and H3K27ac and in DNase accessibility in *MYBPH* (here reported as *MYOG/MYBPH*), *MEF2A* and *TNNI1* enhancers (Ancillary Figure 5 for the Reviewer). Of note, the *MYBPH* enhancer region is reported in Figure 5f (Hi-ChIP) as looping on the *MYOG* promoter. On the *RYR1* regulatory region, loss of *SNAI2* binding leads to a significant increase in DNase accessibility despite any evident increase in MYOD occupancy and H3K27ac binding after Trametinib (Ancillary Figure 5 for the Reviewer). Interestingly, the *MYOG* promoter regulatory region characterized by a very slight binding and reduction of *SNAI2* (Ancillary Figure 5 for Reviewer) shows an enhancement of DNase accessibility in the absence of MYOD and H3K27ac increase. This could suggest that the accessibility to DNA is not regulated by the pure presence of the muscle master gene and H3K27ac binding but can depend on other factors among which the formation of new transcriptional complexes and/or the looping of enhancers.

Minor concerns:

1. In Fig 3P, the error bar for the second shScr group is missing.

We have corrected this oversight.

2. In Fig S3S, the numbers of mice in each group vary greatly, what is the rationale for that design?

Comment for Reviewer 1 and 2. In our initial submission we injected RD cells in 6 shScr mice on the left leg and 3 of them with either shSNAI2-1 or shSNAI2-2 in the right leg to reduce the number of mice having, at the same time, sufficient power calculation (6 control samples and 6 SNAI2-depleted samples). However, we do agree with the reviewer that for each shRNA, 3 shSNAI2 xenografted mice used though significant is underpowered; hence we provide new *in vivo* experiments in SMS-CTR cells (Figure S5 f-m).

Our initial expectation was to observe a significant difference between Scrambled controls and SNAI2 knockdown based on our *in vitro* experiments and that the vincristine treatment could have only a small additive effect. We therefore hypothesized that, to see the difference, 10 mice per shRNA group treated with vincristine, with 2 tumors per mouse would be sufficient to observe a difference in growth. However, our unexpected results show that SNAI2 ablation in combination with standard of care vincristine greatly reduces tumor size and also significantly enhances tumor differentiation. We would like to explore in future experiments the interaction between SNAI2 ablation and vincristine treatment.

3. The colors are too similar to be distinguished among groups in Fig 4D and 5D.

We made the change as requested to clearly demarcate groups in the analyses.

Reviewer #3 (Remarks to the Author):

The paper by Pomella et al identifies the transcription factor SNAI2 as oncogene in fusion-negative rhabdomyosarcoma. It is convincingly shown that reduction of SNAI2 levels induces differentiation both *in vitro* as well as *in vivo* by a variety of cell biological assays in three different cell lines. Mechanistically, the authors identify enhancers bound by SNAI2 by ChIPseq experiments acting in competition to MYOD binding at E-boxes of differentiation genes. In addition, SNAI2 suppresses the myogenic program via down-regulation of MYOG, MEF2A and CDKN1A. The findings are novel and advance our understanding of the differentiation block found in rhabdomyosarcoma despite expression of myogenic master regulators.

Minor points:

1) The abstract lacks a concluding sentence.

We thank the reviewer for this suggestion and have added a concluding sentence.

2) Figure 1B: SNAI2 expression in Rh18 cells is very low.
Does this also result in a differentiation block?

We also assessed differentiation in additional FN-RMS RH18 and SMS-CTR cells and find a significant and robust increase in myogenic differentiation similar to all other FN-ERMS cell lines tested (Ancillary Figure 4I-L, Figure S2 j-m, Figure S3 g, l).

3) Supplemental Figure 1C lacks statistics

This figure is re-annotated as Figure S1 e. We were unable to perform statistics since this was based on reanalysis of public data (GSE93263) which included only useful duplicates. In addition, the basal level of SNAI2 was high in the parental cell line, but we nevertheless see an increase upon forced expression of *MYOD1*.

4) Supplemental Figure 2H, what are RD18 cells? They are not mentioned in Materials.

We updated the Methods section to include the information. RD18 are a sub clone of RD cells (Ref. 43)

5) Figure 4, S4 and following: In Figure S4, the authors demonstrate that only 146 SNAI2 binding peaks are common to all three cell lines analyzed. It is unclear whether go analysis in 4B refers to these overlapping sites/genes or not. If not, a GO analysis specifically for these overlaps should be shown. Further, Figure 4 talks about 664 sites overlapping between SNAI2 and MYOD in all 3 cell lines (legend). These numbers are not consistent.

To clarify, we performed ChIPseq in 3 FN-RMS cell lines SMS-CTR, RD and JR1 comparing Scrambled shRNA control to SNAI2 shRNA knockdown cells. There are 1,069 peaks that appear in any two of the three cell lines tested; these are the ones used for the GO enrichment. The 146 sites present in all three are too few to enrich for GO terms. The sites with peaks called in only 2 out of 3 still have signal present in the cell line where it isn't called, but just a little too weak to meet our peak calling stringency criteria. We feel confident that our 1,069 peaks are the correct set to use for all the analysis we have done.

The 664 sites of SNAI2 and MYOD overlap is the subset of the 1069 peaks while the rest (405) was considered SNAI2 only binding sites.

6) Figure 5C: Differences in MYOD ChIPseq at MYOD sites appear very minor, Is this really significant?

We agree with the reviewer that the changes are minor. However, the direction of change is different depending on the presence or absence of SNAI2: a minor increase in MYOD binding is seen at sites where SNAI2 was overlapping, but a minor decrease in MYOD binding at sites lacking SNAI2.

To address this further, we zoomed in on the sites that change the most (up or down) after SNAI2 knockout (Figure S7 a, b). The distribution of MYOD change was clearly different at sites overlapping SNAI2 (Figure S7 a). A Fisher's exact test demonstrated significance for overlap with SNAI2 at sites with the greatest MYOD increase (above what is expected by random chance, Figure S7 b).

7) Figure 6A: The scheme should not have an activating arrow from "SNAI2" to "myogenic TF activated", as is rather blocks differentiation.

We thank the reviewer for catching this error and have changed our graphic to show that SNAI2 represses myogenic differentiation and once SNAI2 is ablated, FN-RMS tumors can undergo terminal differentiation.

Ancillary Figure 1

Ancillary Figure 1 related to Figure 2; Suppression of *SNAI2* affects stemness gene expression in FN-RMS.

(A) qRT-PCR gene expression analysis of RD cell spheres comparing stemness markers in shScr and shSNAI2.1 knock-down after 15 days in sphere formation medium (Neurobasal medium).

(B) GSEA analysis of gene expression changes associated with shSNAI2.1 in RD spheres. RD cells containing shScr or shSNAI2.1 after 15 days in neurobasal medium were analyzed by RNA-seq, the expression dataset was evaluated against published embryonic stem cell signature from GSEA dataset (Bhattacharya et.al 2004). Similar to the observation made by qRT-PCR, embryonic stem cell associated markers were downregulated in RD spheres with shSNAI2 KD.

Ancillary Figure 2

Ancillary Figure 2 related to Figure 2; *MEF2C* and *CDKN1A* are indirect targets of SNAI2.

Tracks of SNAI2, MYOD, H3K27ac ChIP-seq shown at *MEF2C* and *CDKN1A* upon shSNAI2 compared to shScramble.

Ancillary Figure 3

Ancillary Figure 3 related to Figure S1; *SNAI2* is highly expressed in RMS and is regulated by MYOD.

(A) A validated siRNA targeting MYOD was used to knockdown MYOD expression in RD FN-RMS cells. MYOD and SNAI2 expression was detected by western blot. Blots of biological triplicates are reported (n=1; n=2; n=3).

n=1 reloaded blot (second blot from left) shows MYOD and SNAI2 staining on the same samples used in n=1 (first blot from left) that was present in the previous submitted Figure 1h. This new reloaded blot shows now comparable levels of MYOD in siScr at 24 and 48h, highlighting a possible technical issue on MYOD staining in siScr at 24h in the previous reported western blot. Red box highlights reversely loaded samples for siScr and siMYOD.

(B) CRISPR-Cas9 targeting *MYOD1* was used to knockout MYOD expression in RD cells. MYOD and SNAI2 expression was detected by western blot (left) and qRT-PCR (right). Data was normalized to cells treated with non-targeting sgRNA (sgNT). Bars show median (error bars = range) of triplicates of a single experiment performed at 48 and 72 hours post sgRNA transfection.

(C,D,E) A validated siRNA targeting MYOD was used to knockdown MYOD expression in RH30, SCMC, RH4 FP-RMS cells. MYOD and SNAI2 expression was detected by western blot (left) and qRT-PCR (right). Red box in **E** highlights samples were loaded in reversed order for siScr and siMYOD.

Data was normalized to cells treated with scramble siRNA. Bars show median (error bars = range) of 3 independent experiments performed at 24 and 48 hours of transient siRNA transfection. Statistical significance was assessed by paired Student's T-Test. (p-value * ≤ 0.05 , ** ≤ 0.001 , *** ≤ 0.0001).

Ancillary Figure 4

Ancillary Figure 4 related to Figure 2; Suppression of *SNAI2* activates myogenic differentiation in FP-RMS and FN-RMS.

(A,E,I) Level of *SNAI2* knockdown by shSNAI2.1 and shSNAI2.2 in RH30 and RH41 FP-RMS cells and RH18 FN-RMS compared to shScr assessed by western blot.

(B-C, F-G, J-K) Immunostaining of RH30, RH41 and RH18 cells stably expressing shScr or shSNAI2.1 shRNA, stained for Myosin Heavy Chain 1 (MyHC, green), MEF2C (red) and DAPI for nuclei (blue).

(D,H,L) Quantitation of immunostaining counts of RH30, RH41 and RH18 cells as percentage value to total nuclei per image (averaged over three images).

Ancillary Figure 5

Ancillary Figure 5 related to Figure 5; DNase accessibility of SNAI2 bound regions are affected upon Trametinib treatment.

DNase accessibility at SNAI2 binding sites shown at representative regions of *MYOG/MYBPH*, *MEF2A* enhancer, *TNNI1*, *RYR1* and *MYOG* promoter. Tracks of SNAI2, MYOD, H3K27ac and DNase accessible regions are shown upon 72h Trametinib treatment compared to DMSO control.

REVIEWERS' COMMENTS

Reviewer #1 (Remarks to the Author):

In this revised manuscript, the authors have addressed my comments and questions well. This is now an excellent paper. I would like to thank the authors for their very thorough response to each critique raised by each of our reviews. I would also like to congratulate them on a beautiful piece of work.

Reviewer #2 (Remarks to the Author):

The authors have done a thorough job in addressing previous concerns. The current version is significantly improved. No further comments.